# GWAS meta-analysis of over 29,000 people with epilepsy identifies 26 risk loci and subtype-specific genetic architecture

**International League Against Epilepsy Consortium on Complex Epilepsies***

Epilepsy is a highly heritable disorder affecting over 50 million people worldwide, of which about one-third are resistant to current treatments. Here we report a multi-ancestry genome-wide association study including 29,944 cases, stratified into three broad categories and seven subtypes of epilepsy, and 52,538 controls. We identify 26 genome-wide significant loci, 19 of which are specific to genetic generalized epilepsy (GGE). We implicate 29 likely causal genes underlying these 26 loci. SNP-based heritability analyses show that common variants explain between 39.6% and 90% of genetic risk for GGE and its subtypes. Subtype analysis revealed markedly different genetic architectures between focal and generalized epilepsies. Gene-set analyses of GGE signals implicate synaptic processes in both excitatory and inhibitory neurons in the brain. Prioritized candidate genes overlap with monogenic epilepsy genes and with targets of current antiseizure medications. Finally, we leverage our results to identify alternate drugs with predicted efficacy if repurposed for epilepsy treatment.

The epilepsies are a heterogeneous group of neurological disorders, characterized by an enduring predisposition to generate unprovoked seizures[1]. It is estimated that over 50 million people worldwide have active epilepsy, with an annual cumulative incidence of 68 per 100,000 persons[2].

Similar to other common neurodevelopmental disorders, epilepsies have substantial genetic risk contributions from both common and rare genetic variations. Analysis of the epilepsies benefits from deep phenotyping, which allows clinical subtypes to be distinguished[3], in contrast to other common neurodevelopmental disorders, where phenotypic subtypes are more difficult to define. Differences in the genetic architecture of clinical subtypes of epilepsy are also emerging, to complement the clinical partitioning[4–7]. The rare but severe epileptic encephalopathies are usually nonfamilial and are largely caused by single de novo dominant variants, often involving genes encoding ion channels or proteins of the synaptic machinery[8]. Both common and rare variants have been shown to contribute to the milder and more common focal and generalized epilepsies. This is particularly true for generalized epilepsy, which is primarily constituted by genetic generalized epilepsy (GGE)[4,5,9,10]. Nevertheless, previous genetic studies of common epilepsies have explained only a limited proportion of this common genetic variant, or single-nucleotide polymorphism (SNP)-based, heritability—9.2% for focal and 32.1% for GGE[4–6,10].

Epilepsy is typically treated using antiseizure medications (ASMs). However, despite the availability of over 25 licensed ASMs worldwide, a third of people with epilepsy experience continuing seizures[11]. Diet, surgery and neuromodulation represent additional treatment options that can be effective in small subgroups of patients[12]. Accurate classification of clinical presentations is an important guiding factor in epilepsy treatment.

Here we report the third epilepsy genome-wide association study (GWAS) meta-analysis by the International League against Epilepsy (ILAE) Consortium on complex epilepsies, comprising a total of 29,944 deeply phenotyped cases recruited from tertiary referral centers and 52,538 controls, approximately doubling the previous sample size[4]. Results suggest markedly different genetic architectures between focal and generalized forms of epilepsy. Combining these results with those from less-stringently phenotyped biobank

and deCODE genetics epilepsy cases did not substantially increase signal, despite almost doubling the sample size to 51,678 cases and 1,076,527 controls. Our findings shed light on the enigmatic biology of generalized epilepsy and the importance of accurate syndromic phenotyping and may facilitate drug repurposing for new therapeutic approaches.

## Results

### Study overview

We performed a GWAS meta-analysis by combining the previously published effort from our consortium[4] with unpublished data from the Epi25 collaborative[10] and four additional cohorts (Supplementary Tables 1 and 2). Our primary mixed model meta-analysis constitutes 4.9 million SNPs tested in 52,538 controls and 29,944 people with epilepsy, of which 16,384 had neurologist-classified focal epilepsy (FE) and 7,407 had GGE. The epilepsy cases were primarily of European descent (92%), with a smaller proportion of African (3%) and Asian (5%) ancestry (Supplementary Table 3). Cases were matched with controls of the same ancestry, and GWAS analyses were performed separately per ancestry, before performing multi-ancestry meta-analyses for the broad epilepsy phenotypes 'FE' ($n = 16,384$ cases) and 'GGE' ($n = 7,407$ cases). We further conducted meta-analyses in individuals of European ancestry of the well-defined GGE subtypes of juvenile myoclonic epilepsy (JME; $n = 1,732$), childhood absence epilepsy (CAE; $n = 1,049$), juvenile absence epilepsy (JAE; $n = 662$) and generalized tonic-clonic seizures alone (GTCSA; $n = 485$), as well as the FE subtypes of FE with hippocampal sclerosis (HS; $n = 1,260$), FE with other lesions ($n = 4,213$) and lesion-negative FE ($n = 5,778$). The same controls ($n = 42,436$) were shared across the different subphenotypes. We ran a variety of follow-up analyses to identify potential sex-specific signals and obtain biological insights and opportunities for drug repurposing. Sample size prevented the inclusion of other ethnicities in the subtype analyses.

### GWAS for the epilepsies

Our 'all epilepsy' meta-analysis revealed four genome-wide significant loci, of which two are new (Fig. 1). Similar to our previous GWAS[4], the 2q24.3 locus was composed of two independently significant signals (Supplementary Table 4). Using ASSET to determine the extent of FE and GGE-related pleiotropy, the 2q24.3 and 9q21.13 signals showed pleiotropic effects at a genome-wide significance level, with concordant SNP effect directions for both forms of epilepsy (Supplementary Table 5). The 2p16.1 and 10q24.32 loci were primarily derived from GGE. The FE analysis did not reveal any genome-wide significant signals.

Our 'GGE' meta-analysis uncovered a total of 25 independent genome-wide significant signals across 22 loci, of which 13 loci are new. The strongest signal of association ($P = 6.6 \times 10^{-21}$), located at 2p16.1, constitutes three independently significant signals. Similarly, the new locus 12q13.13 was composed of two independently significant signals (Supplementary Table 4). Forest plots and P–M plots of these signals show that they appear consistent across all four GGE subphenotypes, with some exceptions (Supplementary Figs. 1 and 2).

We applied multitrait analysis of GWAS (MTAG)[17] to exploit the correlation between FE and GGE, boosting the effective sample size. Results were concordant with our main analysis, and new signals did not emerge (Supplementary Fig. 3).

Functional annotation of the 1,082 genome-wide significant SNPs across the 22 GGE loci and 270 SNPs from the 'all epilepsy' loci revealed that most variants were intergenic or intronic (Supplementary Data 1). Eight of 1,082 (0.7%) GGE SNPs were exonic, of which five were located in protein-coding genes and were missense variants. We identified one exonic 'all epilepsy' SNP (rs7580482, synonymous), located in *SCN1A*. Seventy-four percent of 'all epilepsy' SNPs and 64% of GGE SNPs were located in open chromatin regions, as indicated by a minimum chromatin state of 1–7 (ref. 14). Further annotation by Combined

Annotation-Dependent Depletion (CADD) scores predicted that 11 'all epilepsy' and 50 GGE SNPs were deleterious (CADD score > 12.37) (ref. 15). LDAK heritability analyses showed significant enrichment of signal in 'super-enhancers' (Supplementary Table 6), suggesting that GGE SNPs regulate clusters of transcriptional enhancers that control the expression of genes that define cell identity[16].

To assess potential syndrome-specific loci, we performed GWAS on seven well-defined FE and GGE subtypes (Supplementary Fig. 4a–g). We found three genome-wide significant loci associated specifically with JME ($n = 1,813$), of which one was new (8q23.1) and the other two (4p12 and 16p11.2) previously reported[4]. Our analysis of CAE ($n = 1,072$) consolidated an established genome-wide significant signal at 2p16.1, which was also observed in the GGE and all epilepsy GWAS. We did not find any genome-wide significant loci for JAE ($n = 671$), GTCSA ($n = 499$), 'nonlesional FE' ($n = 6,367$), 'FE with HS' ($n = 1,375$) or 'FE with other lesions' ($n = 4,661$).

MTAG[17] analysis of individual GGE subphenotypes showed concordance with the main GGE GWAS, without identifying new loci. In addition, this analysis confirmed that the majority of GWAS-significant SNPs in GGE are overlapping (Supplementary Figs. 5 and 6 and Supplementary Table 7).

The vast majority of loci reported in our previous effort[4] remained genome-wide significant. A summary of loci that fell below the genome-wide significance threshold is provided in Supplementary Table 8.

Genomic inflation was comparable to our previous GWAS, and all linkage-disequilibrium score regression (LDSC) intercepts were lower (Supplementary Table 9)[4], suggesting that the signals are primarily driven by polygenicity. Computation of the attenuation ratio suggested that part of the inflation signal, in particular for FE (0.58), might be due to some form of bias (for example, confounding or population stratification)[13]. The attenuation ratio was lowest for GGE (0.11), which includes the vast majority of significant loci (Supplementary Table 9).

### Locus annotation, gene-based analyses and gene prioritization

Using FUMA[18] (Methods), the 'all epilepsy' meta-analysis was mapped to 43 genes and the GGE analysis to 278 genes (Supplementary Data 2). Thirty-nine of the 43 'all epilepsy' genes overlapped with GGE, resulting in a total of 282 uniquely mapped genes. These 282 genes were enriched for monogenic epilepsy genes (hypergeometric test, 18/837 genes overlapped; odds ratio (OR) = 1.51, $P = 0.04$) and targets of ASMs (hypergeometric test, 9/191 genes overlap; OR = 3.39, $P = 5.4 \times 10^{-4}$).

We calculated a gene-based association score based on the aggregate of all SNPs inside each gene using MAGMA (Methods)[19]. This analysis yielded 39 significant genic associations—six with 'all epilepsy' and 37 with GGE (four overlapped with the 'all epilepsy' analysis), after correction for 16,371 tested genes ($P < 0.05/16,371$ genes; Supplementary Data 3). Thirteen of these 39 genes mapped to regions outside of the genome-wide significant loci from the single SNP analyses.

Next, we performed a transcriptome-wide association study (TWAS) to assess whether epilepsy was associated with differential gene expression in the brain (Methods)[20,21]. These analyses revealed significant associations with 27 genes in total; 13 genes with 'all epilepsy,' 16 with GGE and two with both phenotypes (Supplementary Data 4). Nineteen of the 27 genes mapped outside of the 26 loci were identified through the GWAS. Using summary-data-based Mendelian randomization (SMR)[22], we determined a potentially causal relationship between brain expression of *RMI1* and 'all epilepsy,' and among *RMI1*, *CDK5RAP3* and *TVP23B* and GGE (Supplementary Data 5).

Of note, expression of *RMI1* was associated with GGE in both TWAS ($P = 4.0 \times 10^{-10}$) and SMR ($P = 5.2 \times 10^{-8}$), as well as with 'all epilepsy' (TWAS $P = 1.3 \times 10^{-6}$; SMR $P = 2.6 \times 10^{-6}$). *RMI1* has a crucial role in genomic stability[23] and has not been previously associated with epilepsy or any other Mendelian trait (OMIM, 610404).

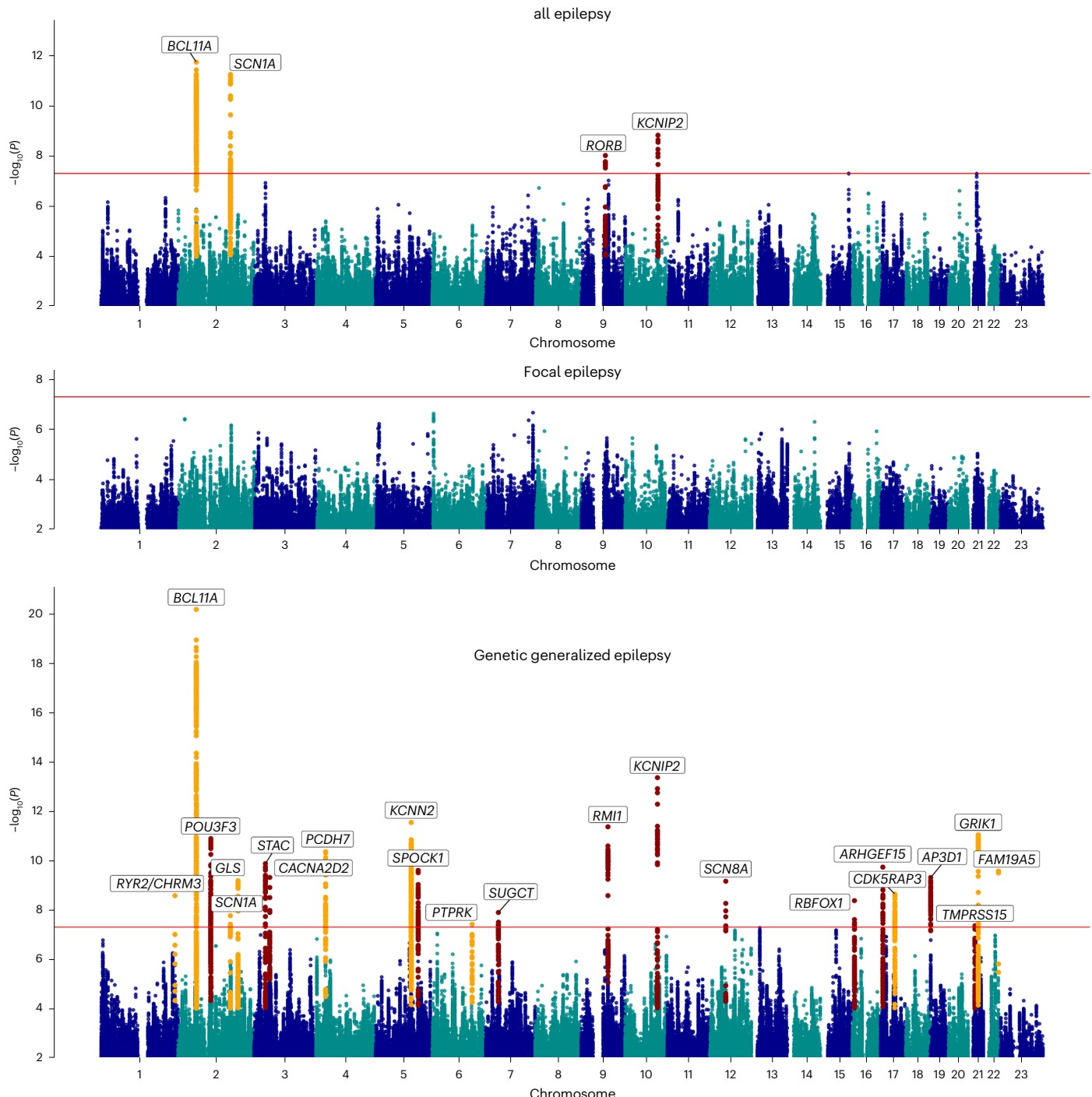

**Fig. 1 | Manhattan plot of multi-ancestry all epilepsy ($n$ = 29,944), focal epilepsy ($n$ = 16,384) and genetic generalized epilepsy ($n$ = 7,407) genome-wide meta-analyses, obtained by fixed-effects meta-analysis weighted by effective sample sizes.** The red line shows the genome-wide significance threshold ($5 \times 10^{-8}$). Chromosome and position are displayed on the $x$ axis, and two-sided $-\log_{10} P$ value is on the $y$ axis. New genome-wide significant loci are highlighted in red, and loci previously associated with epilepsy in orange. New loci were those previously unreported as GWAS significant in previous epilepsy GWASs. Annotated genes are those implicated by our gene prioritization analyses. See Supplementary Fig. 7 for QQ plots. QQ plots, quantile–quantile plot.

We used a combination of ten different criteria to identify the most likely implicated gene within each of the 26 associated loci from the meta-analysis (Methods). This resulted in a shortlist of 29 genes (Table 1; see Supplementary Data 6 for scores of all mapped genes), of which ten are monogenic epilepsy genes, seven are known targets of currently licensed ASDs and 17 are associated with epilepsy for the first time.

The strongest association signal for GGE was found at 2p16.1, consistent with our previous results where we implicated *VRK2* or

*FANCL*[24]. Our gene prioritization analysis suggests the transcription factor *BCL11A* as the culprit gene, located 2.5 Mb upstream of the lead SNPs at this locus. Two of three lead SNPs are in enhancer regions (as assessed by chromatin states in brain tissue) that are linked to the *BCL11A* promoter via 3D chromatin interactions (Supplementary Fig. 8). Rare variants in *BCL11A* were recently associated with intellectual disability and epileptic encephalopathy[25]. However, interrogation of the MetaBrain expression quantitative trait loci (eQTL)

**Table 1 | Genome-wide significant loci and prioritized genes**

| Phenotype | Locus | New/replication | Lead SNP (A1:A2) | Freq1 | Z score | P value | Genes | Total | Missense | TWAS | SMR | MAGMA | PoPS | Brain exp | Brain-coX | KO mouse | AED target | Monogenic |
|---|---|---|---|---|---|---|---|---|---|---|---|---|---|---|---|---|---|---|
| All epilepsy | 2p16.1 | Replication | rs13032423 (A:G) | 0.53 | −7.04 | $1.85 \times 10^{-12}$ | BCL11A | 5 | − | − | − | − | * | * | * | * | − | * |
| | 2q24.3 | Replication | rs59237858 (T:C) | 0.23 | −6.89 | $5.75 \times 10^{-12}$ | SCN1A | 8 | * | − | − | * | * | * | * | * | * | * |
| | 9q21.13 | New | rs4744696 (A:G) | 0.82 | −5.74 | $9.69 \times 10^{-9}$ | RORB | 4 | − | − | − | − | * | * | * | * | − | − |
| | 10q24.32 | New | rs3740422 (C:G) | 0.33 | 6.04 | $1.52 \times 10^{-9}$ | KCNIP2 | 3 | − | − | − | * | * | * | * | − | − | − |
| | 1q43 | New | rs876793 (T:C) | 0.67 | −5.95 | $2.64 \times 10^{-9}$ | RYR2 | 4 | − | − | − | − | * | * | * | * | − | − |
| | | | | | | | CHRM3 | 4 | − | − | − | − | − | * | * | * | * | − |
| | 2p16.1 | Replication | rs11688767 (A:T) | 0.53 | 9.38 | $6.58 \times 10^{-21}$ | BCL11A | 5 | − | − | − | − | * | * | * | * | − | * |
| | 2q12.1 | New | rs62151809 (T:C) | 0.43 | 6.77 | $1.28 \times 10^{-11}$ | POU3F3 | 3 | − | − | − | − | * | * | − | * | − | − |
| | 2q24.3 | Replication | rs11890028 (T:G) | 0.72 | 5.63 | $1.73 \times 10^{-8}$ | SCN1A | 8 | * | − | − | * | * | * | * | * | * | * |
| | 2q32.2 | Replication | rs6721964 (A:G) | 0.66 | −6.18 | $6.54 \times 10^{-10}$ | GLS | 4 | − | − | − | − | * | * | * | * | − | * |
| | 3p22.3 | New | rs9861238 (A:G) | 0.41 | −6.42 | $1.33 \times 10^{-10}$ | STAC | 2 | − | − | − | − | * | − | * | − | − | − |
| | 3p21.31 | New | rs739431 (A:G) | 0.84 | 6.23 | $4.82 \times 10^{-10}$ | CACNA2D2 | 6 | − | − | − | * | − | * | * | * | * | − |
| | 4p15.1 | Replication | rs1463849 (A:G) | 0.59 | −6.59 | $4.38 \times 10^{-11}$ | PCDH7 | 3 | − | − | − | * | * | − | * | − | − | − |
| | 5q22.3 | Replication | rs4596374 (T:C) | 0.55 | −6.98 | $2.91 \times 10^{-12}$ | KCNN2 | 6 | − | − | − | * | * | * | * | * | − | − |
| | 5q31.2 | New | rs2905552 (C:G) | 0.48 | −6.33 | $2.49 \times 10^{-10}$ | SPOCK1 | 5 | * | − | − | * | * | * | * | * | − | − |
| | 6q22.33 | Replication | rs13219424 (T:C) | 0.29 | −5.49 | $3.87 \times 10^{-8}$ | PTPRK | 3 | − | − | − | * | * | − | * | − | − | − |
| | 7p14.1 | New | rs37276 (T:G) | 0.26 | −5.69 | $1.29 \times 10^{-8}$ | SUGCT | 2 | − | * | − | − | * | * | − | * | − | − |
| | 9q21.32 | New | rs2780103 (T:C) | 0.26 | −6.93 | $4.34 \times 10^{-12}$ | RMI1 | 5 | * | * | * | * | − | * | − | * | − | − |
| | 10q24.32 | New | rs11191156 (A:G) | 0.67 | −7.55 | $4.41 \times 10^{-14}$ | KCNIP2 | 4 | − | − | − | * | * | * | * | * | − | − |
| | 12q13.13 | New | rs114131287 (A:T) | 0.02 | 5.83 | $5.46 \times 10^{-9}$ | SCN8A | 6 | − | − | − | − | * | * | * | * | * | * |
| | 16p13.3 | New | rs62014006 (T:G) | 0.05 | 5.88 | $4.22 \times 10^{-9}$ | RBFOX1 | 5 | − | − | − | * | * | * | * | * | − | − |
| | 17p13.1 | New | rs2585398 (A:C) | 0.53 | −6.37 | $1.84 \times 10^{-10}$ | ARHGEF15 | 6 | * | * | * | * | − | * | * | * | − | − |
| | 17q21.32 | Replication | rs16955463 (T:G) | 0.25 | −5.97 | $2.30 \times 10^{-9}$ | CDK5RAP3 | 4 | − | * | * | * | − | − | − | * | − | − |
| | 19p13.3 | New | rs75483641 (T:C) | 0.14 | −6.22 | $4.85 \times 10^{-10}$ | AP3D1 | 5 | * | − | * | * | * | − | − | * | − | − |
| | 21q21.1 | New | rs1487946 (A:G) | 0.59 | 5.47 | $4.41 \times 10^{-8}$ | TMPRSS15 | 1 | − | − | − | − | * | − | − | − | − | − |
| | 21q22.1 | Replication | rs7277479 (A:G) | 0.36 | −6.82 | $8.94 \times 10^{-12}$ | GRIK1 | 4 | − | − | − | − | * | * | * | * | * | − |
| | 22q13.32 | New | rs469999 (A:G) | 0.31 | −6.32 | $2.65 \times 10^{-10}$ | FAMI9A5 | 2 | − | − | − | * | * | * | − | − | − | − |
| GGE | | | | | | | | | | | | | | | | | | |
| CAE | 2p16.1 | Replication | rs12185644 (A:C) | 0.70 | −7.12 | $1.04 \times 10^{-12}$ | BCL11A | 5 | − | − | − | − | * | * | * | * | − | * |
| JME | 4p12 | Replication | rs17537141 (T:C) | 0.851 | −5.47 | $4.62 \times 10^{-8}$ | GABRA2 | 6 | − | − | − | * | − | * | * | * | * | * |
| | 8q23.1 | New | rs3019359 (T:C) | 0.414 | −5.55 | $2.89 \times 10^{-8}$ | RSPO2 | 3 | − | − | − | − | * | * | * | − | − | − |
| | | | | | | | TMEM74 | 3 | − | − | − | − | − | * | * | * | − | − |
| | 16p11.2 | Replication | rs1046276 (T:C) | 0.353 | 6.19 | $6.05 \times 10^{-10}$ | STX1B | 5 | − | * | − | * | − | * | * | * | * | * |
| | | | | | | | CACNA1I | 5 | − | − | − | − | − | * | * | * | * | * |

Genome-wide significant loci are annotated with details from the lead-SNP and prioritized genes. Loci were classified as new or replication according to the genome-wide significant results of previous GWAS publications. Genes were scored based on ten criteria/methods, after which the gene with the highest score in the locus was selected as the prioritized gene. Genomic coordinates for each locus (hg19) can be found in Supplementary Table 4. Two-tailed P values and z scores were obtained by fixed-effects meta-analysis weighted by effective sample sizes. Total, number of satisfied criteria for gene prioritization; missense, the locus contains a missense variant in the gene; TWAS, significant transcriptome-wide association with the gene; SMR, significant summary-based Mendelian randomization association with the gene; MAGMA, significant genome-wide gene-based association; PoPS, gene prioritized by polygenic priority score; brain exp, the gene is preferentially expressed in brain tissue; brain-coX, the gene is prioritized as co-expressed with established epilepsy genes; KO mouse, knockout of the gene causes a neurological phenotype in mouse models; monogenic, the gene is a known cause of monogenic epilepsy.

database did not reveal a significant association of our lead SNPs with *BCL11A* expression.

## The HLA system and common epilepsies

The highly polymorphic HLA region has been associated with various neuropsychiatric and autoimmune neurological disorders. Therefore, we imputed HLA alleles and amino acid residues using CookHLA v1.0.1 (ref. [26]) and ran association across epilepsy, focal and GGE phenotypes, as well as the seven subphenotypes (Methods). No SNP, amino acid residue or HLA allele reached genome-wide significance (Supplementary Fig. 9). The most significant signal was an aspartame amino acid residue in exon 2 of *HLA-B* (position 31432494), which had a *P* value of $3.8 \times 10^{-7}$ for GGE.

## SNP-based heritability

We calculated SNP-based heritability using LDAK to determine the proportion of epilepsy risk attributable to common genetic variants. We observed liability scale SNP-based heritabilities of 17.7% (95% confidence interval (CI): 15.5–19.9%) for all epilepsy, 16.0% (14.0–18.0%) for FE and 39.6% (34.3–44.6%) for GGE. Heritabilities were notably higher for all individual GGE subtypes, ranging from 49.6% (14.0–85.3%) for GTCSA to 90.0% (63.3–116.6%) for JAE (Supplementary Table 10).

Using a univariate causal mixture model[27] (Methods), we estimated that 2,850 causal SNPs (s.e.: 200) underlie 90% of the SNP-based heritability of GGE, comparable with previous estimates[9]. Power analysis demonstrated that the current genome-wide significant SNPs only explain 1.5% of the phenotypic variance, whereas an estimated sample size of around 2.5 million individuals would be necessary to identify the causal SNPs that explain 90% of GGE SNP-based heritability (Supplementary Fig. 10).

To further explore the heritability of the different epilepsy phenotypes, we used LDSC to perform genetic correlation analyses[28]. We found evidence for a strong genetic correlation among all four GGE syndromes (Supplementary Fig. 11 and Supplementary Table 11). We also observed the previously reported significant genetic correlation[4] between the focal nonlesional and JME syndromes. Here CAE also showed a significant genetic correlation with the focal nonlesional cohort. Multivariate modeling of genetic correlation using Genomic structural equation modeling (SEM)[29] confirmed that most of the heritability signal is shared among the four GGE syndromes, with some subtype-specific signals (Supplementary Fig. 12).

## Tissue and cell type enrichment

To further illuminate the underlying biological causes of the epilepsies, we used MAGMA[19] and data from the gene–tissue expression (GTEx) consortium to assess whether our GGE-associated genes were enriched for expression in specific tissues and cell types (Methods). We identified significant enrichment of associated genes expressed in brain and pituitary tissue (Supplementary Fig. 13). The implication of the pituitary gland in GGE might reflect a hormonal component to seizure susceptibility. Further subanalyses showed that our results were enriched for genes expressed in almost all brain regions, including subcortical structures such as the hypothalamus, hippocampus and amygdala (Supplementary Fig. 14). We did not find enrichment for genes expressed at specific developmental stages in the brain (Supplementary Fig. 15).

Cell-type specificity analyses of GGE data using various single-cell RNA-sequencing reference datasets (Methods) revealed enrichment in excitatory as well as inhibitory neurons, but not in other brain cells like astrocytes, oligodendrocytes or microglia (Supplementary Fig. 16). Similarly, stratified linkage-disequilibrium (LD)-score regression using single-cell expression data (Methods) did not reveal a difference between excitatory and inhibitory neurons ($P = 0.18$).

## Gene-set analyses

MAGMA gene-set analyses showed significant associations between GGE and biological processes involving various functions in the synapse (Supplementary Data 7). To further refine the synaptic signal, we performed a gene-set analysis using lists of expert-curated gene sets involving 18 different synaptic functions[30]. These analyses showed that GGE was associated with intracellular signal transduction ($n = 139$ genes, $P = 9.6 \times 10^{-5}$) and excitability in the synapse ($n = 54$ genes, $P = 0.0074$). None of the other 16 synaptic functions showed any association (Supplementary Data 7). Genes involved with excitability include the N-type calcium channel gene *CACNA2D2*, implicated at the new GGE locus 3p21.31. N-type calcium channel blockers such as levetiracetam and lamotrigine are among the most widely used and effective ASMs for GGE as well as FE[31–33]. Together, these results suggest that the genes associated with GGE are expressed in excitatory as well as inhibitory neurons in various brain regions, where they affect excitability and intracellular signal transduction at the synapse.

## Sex-specific analyses

There are known sex-related patterns in the epidemiology of epilepsy. Although females have a marginally lower incidence of epilepsy than males, GGE is known to occur more frequently in females[34]. To test whether this sex divergence has a genetic basis, we performed sex-specific GWAS for 'all', GGE and FE (Supplementary Figs. 17–19). These analyses revealed one female-specific genome-wide significant signal at 10q24.32 (lead SNP: rs72845653), containing *KCNIP2*. This locus was also implicated in our main GGE meta-analysis (lead SNP: rs11191156); however, the lead SNPs of these two signals show low allelic correlation ($r^2 = 0.05$; $D' = 0.87$). Interestingly, the direction of effect of this signal is opposite in females and males. This sex difference is further corroborated by significant sex heterogeneity ($P = 1.54 \times 10^{-8}$) and sex-differentiated GWAS ($P = 5.6 \times 10^{-9}$) (ref. [35]). Sex-related differences in transcription levels in human heart have previously been reported for *KCNIP2* (ref. [36]). We did not find any sex-divergent signals for 'all' or FE. These analyses were limited by a reduction in sample size and prone to random fluctuation.

We used LDSC to assess the genetic correlation between male-only and female-only GWAS. The male and female GWAS of 'all epilepsy,' FE and GGE were strongly genetically correlated (all $r_G > 0.9$), and none of these correlations were significantly different from 1 (all $P > 0.05$). These results suggest that, with the exception of the female-specific 10q24.32 signal, the overall genetic basis of common epilepsy appears largely similar between males and females.

## Genetic overlap between epilepsy and other phenotypes

To explore the genetic overlap of epilepsy with other diseases, we first used the GWAS Catalog[37] to cross-reference the 26 genome-wide epilepsy loci with other traits with significant associations ($P < 5 \times 10^{-8}$) for the same SNP, or SNPs in strong LD with our lead SNPs (as detailed in Table 1). This analysis revealed 18 likely pleiotropic loci, with previous associations reported across a variety of traits, the most common being cognitive, sleep, psychiatric, coronary and blood cell-related (Supplementary Fig. 20). The remaining eight loci appear to be specific to epilepsy (3p22.3, 4p12, 5q31.2, 7p14.1, 8q23.1, 9q21.13, 21q21.1 and 21q22.1).

We then performed genetic correlation analyses between 18 selected traits (Supplementary Table 12) and 'all', GGE and FE using LDSC[13]. The selected traits had either, or a combination of, epilepsy as a common comorbidity or pleiotropic loci shared with epilepsy. Significant correlations ($P < 0.05/54 = 0.0009$) were found with febrile seizures, stroke, headache, ADHD, type 2 diabetes and intelligence (Fig. 2).

Genetic correlation analyses assess the aggregate of shared genetic variants associated with two phenotypes. However, genetic correlations can become close to zero when there is inverse directionality of SNP effects between two phenotypes[38]. To explore this further,

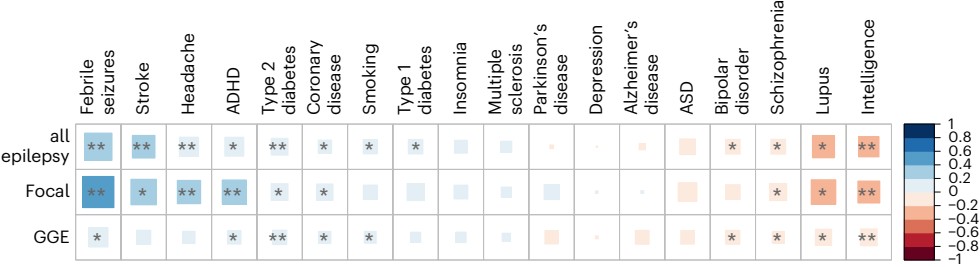

**Fig. 2 | Genetic correlations of epilepsy with other phenotypes.** The genetic correlation coefficient was calculated with LDSC and is denoted by color scale from −1 (red; negatively (anti-)correlated) to +1 (blue; positively correlated). The square size relates to the absolute value of the corresponding correlation coefficient. Single asterisk indicates two-sided $P < 0.05$ and double asterisk indicates two-sided $P < 0.0009$ (Bonferroni corrected).

we applied MiXeR v1.2.0 to quantify the polygenic overlap between GGE and the same 18 selected traits, irrespective of genetic correlation (Methods). Results showed a large polygenic overlap between epilepsy and various other brain traits (Supplementary Fig. 21). For most selected brain traits, the direction of effect was concordant for 40–60% of SNPs. This might explain why some LDSC correlations were low, together with other relevant factors including sample size, polygenicity and trait genetic architecture. In combination, these analyses suggest that the SNPs involved with GGE are highly pleiotropic; a large proportion of the ~2,850 causal SNPs underlying GGE seem to underlie the risk of a wide range of other brain diseases and traits, often with opposing directions of effect. These results emphasize that each phenotype has a specific underlying distribution of effect sizes and directions among shared causal variants, which together explain the shared and unique risk for different brain diseases.

## Leveraging GWAS for drug repurposing

We next tested the potential of our meta-analysis to inform drug repurposing, by predicting the relative efficacy of drugs for epilepsy (Methods). This analysis was based on the predicted ability of each drug to modulate epilepsy-related changes in the function and abundance of proteins, as inferred from the GWAS summary statistics (Methods)[39]. In our predictions for all epilepsy, current ASMs were ranked higher than expected by chance ($P < 1 \times 10^{-6}$) and higher than drugs used to treat any other human disease (Supplementary Data 8). These observations were also true for a 'test set' (randomly selected 50%) of ASMs, when the remaining ASMs ('training set') were used for optimizing the predictions.

For GGE, broad-spectrum ASMs were predicted to be more effective than narrow-spectrum ASMs ($P < 1 \times 10^{-6}$), consistent with clinical experience[40]. Furthermore, the predicted order of efficacy for GGE of individual ASMs matched their observed order in the largest head-to-head randomized controlled clinical trials for generalized epilepsy[33,41], an observation unlikely to occur by chance ($P < 1 \times 10^{-6}$).

Using this approach, we highlight the top 20 drugs that are licensed for conditions other than epilepsy, but are predicted to be efficacious for generalized epilepsy, and additionally have published evidence of antiseizure efficacy from multiple published studies and multiple animal models (Supplementary Table 13). The full list of all predictions can be found in Supplementary Data 9.

## GWAS in epilepsies ascertained from population biobanks

Finally, we leveraged the data from several large-scale population biobanks and from deCODE genetics to explore the consistency of the epilepsy loci in cohorts that were less deeply phenotyped (total cases $n = 21,734$, total controls $n = 1,023,989$, phenotyped using International Classification of Diseases (ICD) codes; Methods; Supplementary Table 14). Forest plots showed a consistent direction of effect between the biobanks and our primary GWAS for all

biobank-genotyped genome-wide significant top SNPs of the 'all epilepsy' GWAS and for all but one GGE top SNP (Supplementary Figs. 22 and 23). Although the biobank and deCODE genetics-specific GWAS did not identify any genome-wide significant loci for GGE or 'all epilepsy,' one significant locus at 2q22.1 (nearest gene, *NXPH2*) emerged for FE (Supplementary Fig. 24).

Meta-analysis of the biobank and deCODE genetics summary statistics with those from the primary epilepsy GWAS identified seven significant loci for the 'all epilepsy' phenotype. Six of these signals were previously identified in the primary 'all epilepsy' ($n = 4$) or the 'GGE' GWAS ($n = 2$). One locus (2q12.1) was new. The combined biobank and deCODE genetics meta-analysis for GGE identified five new loci, but four loci from our primary GWAS fell below the threshold of significance (Supplementary Fig. 25). The combined FE meta-analysis showed no significant associations. LDSC between the biobank/deCODE genetics and the primary GWAS results showed genetic correlations ranging between 0.31 and 0.74 (Supplementary Table 15).

## Discussion

In this study, we leveraged a substantial increase in sample size to uncover 26 common epilepsy risk loci, of which 16 have not been reported previously. Using a combination of ten post-GWAS analysis methods, we pinpointed 29 genes that most likely underlie these signals of association. These signals showed enrichment throughout the brain and indicate an important role for synapse biology in excitatory as well as inhibitory neurons. Drug prioritization from the genetic data highlighted licensed ASMs, ranked the ASMs broadly in line with clinical experience and pointed to drugs for potential repurposing. These findings further our understanding of the pathophysiology of common epilepsies and provide new leads for therapeutics.

The 26 associated loci included some notable monogenic epilepsy genes. These include the calcium channel gene *CACNA2D2*, an established epileptic encephalopathy gene[42] that is directly targeted by ten currently licensed drugs, including two ASMs (gabapentin and pregabalin) as well as the Parkinson's disease drug safinamide and the nonsteroidal anti-inflammatory drug celecoxib. Both safinamide and celecoxib have evidence of antiseizure activity[43,44]. *SCN8A*, which encodes a voltage-gated sodium channel, is an established epileptic encephalopathy gene and is associated here with common epilepsies. Na$_v$1.6 (encoded by *SCN8A*) is targeted by commonly used sodium channel-blocking drugs, the most efficacious ASMs for people with monogenic *SCN8A*-related epilepsies, that are often caused by gain-of-function pathogenic variants[45]. Additional drugs targeting Na$_v$1.6 include safinamide and quinidine. *RYR2* encodes a ryanodine receptor, is an established cardiac disorder gene, has recently been implicated in epilepsy[46,47] and is targeted by caffeine as well simvastatin, atorvastatin and carvedilol. The acetylcholine receptor gene *CHRM3* has been previously associated with epilepsy[48] and is targeted by drugs including solifenacin, used to treat urinary incontinence.

We found that GGE, in particular, has a strong contribution from common genetic variation. When analyzing individual GGE syndromes, we found that up to 90% of liability is attributable to common variants in the JAE subtype, making it among the highest of over 700 traits reported in a large GWAS atlas[49] (albeit with relatively large CIs; Supplementary Table 10). The heritability estimates decrease to 40% for the collective GGE phenotype, possibly due to increased heterogeneity from combining syndromes with pleiotropic as well as syndrome-specific risk loci. Although statistical power drastically decreased when assessing specific GGE syndromes, three loci appeared specific to JME. These findings highlight the unique genetic architecture of the subtypes of common epilepsies, which are characterized by a high degree of both shared and syndrome-specific genetic risk.

In contrast to GGE, for FEs, we found only a minor contribution of common variants, with no variant reaching genome-wide significance. It would seem that FEs, as a group, are far more heterogeneous than GGE, lack (common-variation) loci with high effect sizes, have a higher degree of polygenicity and/or have a lower contribution of common heritable risk variation. Our attempt to mitigate this heterogeneity by performing subtype analysis contrasted with the results from GGE, suggesting different genetic architectures, consistent with the experience from studies of common[9] and rare[5] genetic variation and polygenic risk score analyses[6]. There is also emerging evidence for a substantial role of noninherited, somatic mutations in FEs[50].

This work highlights the challenges of working with epilepsy cohorts ascertained through large biobanking initiatives. Accurate classification of epilepsy requires a combination of clinical features, electrophysiology and neuroimaging. Such details were absent from the biobanks we worked with. Rather, phenotypes were generally limited to ICD codes, which are prone to misclassification[51]. Population biobanks are also probably ascertaining milder epilepsies that are responsive to treatment, contrasting with the enrichment for refractory epilepsies at tertiary referral centers.

Moreover, a proportion of adults with epilepsy have an acquired brain lesion, such as stroke, tumors or head trauma. Biobanks typically provide self-reported clinical information and codes from primary care and inpatient hospital care episodes, but not neurological specialist outpatient records that would indicate whether previous brain insults were considered relevant to epilepsy. As a result, the inclusion of the biobank data appeared to introduce more heterogeneity. This contrasts with genetic mapping of other polygenic diseases like type 2 diabetes and migraine, which are relatively easy and reliable to diagnose and classify, resulting in a great increase in GWAS loci when including data from the same biobanks as included in our study[52,53].

We found enrichment of GGE variants in brain-expressed genes, involving excitatory and inhibitory neurons, but not any other brain cell type. This contrasts with other neurological diseases. For example, microglia are involved in Alzheimer's disease[54] and multiple sclerosis[55], whereas migraine does not appear to have brain cell specificity[53]. We further refine this signal by showing the involvement of synapse biology, primarily intracellular signal transduction and synapse excitability. These findings suggest an important role of synaptic processes in excitatory and inhibitory neurons throughout the brain, which could be a potential therapeutic target. Indeed, synaptic vesicle transport is a known target of the ASMs levetiracetam and brivaracetam[56].

We confirmed that our GWAS-identified genes had substantial overlap with monogenic epilepsy genes. A similar convergence of common and rare variant associations has been observed for other neurological neuropsychiatric conditions including schizophrenia[57] and ALS[58]. The genes prioritized in our GWAS signals also overlapped with known targets of current ASMs[4], and we have provided a list of other drugs that directly target these genes. Moreover, using a systems-based approach[39], we highlight drugs that are predicted to be efficacious when repurposed for epilepsy, based on their ability to perturb function and abundance in gene expression. Insights from GWAS of epilepsy have the potential to accelerate the development of new treatments via the identification of promising drug repurposing candidates for clinical trials[59]. We anticipate that follow-up studies of the highlighted drugs in this study could show clinical efficacy in epilepsy treatment.

In summary, these new data reveal markedly different genetic architectures between the milder and more common focal and generalized epilepsies, provide new biological insights to disease etiology and highlight drugs with predicted efficacy when repurposed for epilepsy treatment.

## Online content

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

## International League Against Epilepsy Consortium on Complex Epilepsies

Remi Stevelink[1], Ciarán Campbell[2,3], Siwei Chen[4,5], Bassel Abou-Khalil[6], Oluyomi M. Adesoji[7], Zaid Afawi[8], Elisabetta Amadori[9,10], Alison Anderson[11,12], Joseph Anderson[13], Danielle M. Andrade[14], Grazia Annesi[15], Pauls Auce[16], Andreja Avbersek[17], Melanie Bahlo[18,19,20], Mark D. Baker[21], Ganna Balagura[9,10], Simona Balestrini[17,22], Carmen Barba[23], Karen Barboza[24], Fabrice Bartolomei[25], Thomas Bast[26,27], Larry Baum[28,29], Tobias Baumgartner[30], Betül Baykan[31,32], Nerses Bebek[31,32], Albert J. Becker[33], Felicitas Becker[34], Caitlin A. Bennett[35], Bianca Berghuis[36], Samuel F. Berkovic[35 ✉], Ahmad Beydoun[37], Claudia Bianchini[23], Francesca Bisulli[38,39], Ilan Blatt[8,40], Dheeraj R. Bobbili[41], Ingo Borggraefe[42,43], Christian Bosselmann[44], Vera Braatz[17,22], Jonathan P. Bradfield[45,46], Knut Brockmann[47], Lawrence C. Brody[48], Russell J. Buono[45,49,50], Robyn M. Busch[51,52,53], Hande Caglayan[54], Ellen Campbell[55], Laura Canafoglia[56], Christina Canavati[57], Gregory D. Cascino[58], Barbara Castellotti[59], Claudia B. Catarino[17], Gianpiero L. Cavalleri[2,3 ✉], Felecia Cerrato[60], Francine Chassoux[61], Stacey S. Cherny[28,62], Ching-Lung Cheung[63], Krishna Chinthapalli[17], I-Jun Chou[64], Seo-Kyung Chung[65,66], Claire Churchhouse[4,5,60], Peggy O. Clark[67], Andrew J. Cole[68], Alastair Compston[69], Antonietta Coppola[70], Mahgenn Cosico[71,72], Patrick Cossette[73], John J. Craig[74], Caroline Cusick[60], Mark J. Daly[4,5,60,75], Lea K. Davis[76,77,78,79], Gerrit-Jan de Haan[80], Norman Delanty[2,3,81], Chantal Depondt[82], Philippe Derambure[83], Orrin Devinsky[84], Lidia Di Vito[38], Dennis J. Dlugos[71], Viola Doccini[23], Colin P. Doherty[3,85], Hany El-Naggar[2,3,81], Christian E. Elger[30], Colin A. Ellis[86], Johan G. Eriksson[87], Annika Faucon[88], Yen-Chen A. Feng[4,5,60,89,90], Lisa Ferguson[52], Thomas N. Ferraro[49,91], Lorenzo Ferri[38,39], Martha Feucht[92], Mark Fitzgerald[71,72,86], Beata Fonferko-Shadrach[21], Francesco Fortunato[93], Silvana Franceschetti[94], Andre Franke[95], Jacqueline A. French[96], Elena Freri[97], Monica Gagliardi[98], Antonio Gambardella[93], Eric B. Geller[99], Tania Giangregorio[38], Leif Gjerstad[100], Tracy Glauser[67], Ethan Goldberg[71,72], Alicia Goldman[101], Tiziana Granata[97], David A. Greenberg[102], Renzo Guerrini[23], Namrata Gupta[5], Kevin F. Haas[6], Hakon Hakonarson[45,103], Kerstin Hallmann[30,104], Emadeldin Hassanin[41,105], Manu Hegde[106], Erin L. Heinzen[107,108], Ingo Helbig[71,72,86,95,109,110], Christian Hengsbach[44], Henrike O. Heyne[5,75,111,112], Shinichi Hirose[113], Edouard Hirsch[114], Helle Hjalgrim[115,116], Daniel P. Howrigan[4,5,60], Donald Hucks[76,79], Po-Cheng Hung[64], Michele Iacomino[10], Lukas L. Imbach[117], Yushi Inoue[118], Atsushi Ishii[119], Jennifer Jamnadas-Khoda[17,120], Lara Jehi[52,53], Michael R. Johnson[121], Reetta Kälviäinen[122,123], Yoichiro Kamatani[124], Moien Kanaan[57], Masahiro Kanai[125,126], Anne-Mari Kantanen[122], Bülent Kara[127], Symon M. Kariuki[128,129,130], Dalia Kasperavičiūte[17], Dorothee Kasteleijn-Nolst Trenite[1], Mitsuhiro Kato[131], Josua Kegele[44], Yeşim Kesim[31], Nathalie Khoueiry-Zgheib[132], Chontelle King[133], Heidi E. Kirsch[106], Karl M. Klein[134,135,136,137], Gerhard Kluger[138,139], Susanne Knake[134,137], Robert C. Knowlton[106], Bobby P. C. Koeleman[1 ✉], Amos D. Korczyn[8], Andreas Kouppars[140], Ioanna Kousiappa[140], Roland Krause[41], Martin Krenn[141], Heinz Krestel[135,137,142,143], Ilona Krey[144], Wolfram S. Kunz[30,145], Mitja I. Kurki[4,5,60,75], Gerhard Kurlemann[146], Ruben Kuznicky[147], Patrick Kwan[11,12,148], Angelo Labate[149], Austin Lacey[2,3,81], Dennis Lal[51,52,60], Zied Landoulsi[41], Yu-Lung Lau[150], Stephen Lauxmann[44], Stephanie L. Leech[35], Anna-Elina Lehesjoki[151], Johannes R. Lemke[144], Holger Lerche[44], Gaetan Lesca[152], Costin Leu[17,51,60], Naomi Lewin[71,72], David Lewis-Smith[71,110,153,154], Gloria H.-Y. Li[63,155], Qingqin S. Li[156], Laura Licchetta[38], Kuang-Lin Lin[64], Dick Lindhout[1,80], Tarja Linnankivi[157,158,159], Iscia Lopes-Cendes[160], Daniel H. Lowenstein[106], Colin H. T. Lui[161], Francesca Madia[10], Sigurdur Magnusson[162], Anthony G. Marson[163], Patrick May[41], Christopher M. McGraw[68], Davide Mei[23], James L. Mills[164], Raffaella Minardi[38], Nasir Mirza[163], Rikke S. Møller[115,116], Anne M. Molloy[165], Martino Montomoli[23], Barbara Mostacci[38], Lorenzo Muccioli[39], Hiltrud Muhle[109], Karen Müller-Schlüter[166], Imad M. Najm[52,53], Wassim Nasreddine[37], Benjamin M. Neale[4,5,60], Bernd Neubauer[167], Charles R. J. C. Newton[128,129,130], Markus M. Nöthen[168], Michael Nothnagel[7,169], Peter Nürnberg[7], Terence J. O'Brien[11,12], Yukinori Okada[126,170], Elías Ólafsson[171], Karen L. Oliver[18,19,35], Çiğdem Özkara[172], Aarno Palotie[4,5,60,75], Faith Pangilinan[48], Savvas S. Papacostas[140], Elena Parrini[23], Carlos N. Pato[173], Michele T. Pato[173], Manuela Pendziwiat[95,109], Slavé Petrovski[11,174], William O. Pickrell[21,175], Rebecca Pinsky[176], Tommaso Pippucci[177], Annapurna Poduri[176], Federica Pondrelli[39], Rob H. W. Powell[175], Michael Privitera[178], Annika Rademacher[109], Rodney Radtke[179], Francesca Ragona[97], Sarah Rau[44], Mark I. Rees[66,180], Brigid M. Regan[35], Philipp S. Reif[134,135,137], Sylvain Rhelms[181,182], Antonella Riva[9,10], Felix Rosenow[134,135,137], Philippe Ryvlin[183], Anni Saarela[122,123], Lynette G. Sadleir[133], Josemir W. Sander[17,22,80], Thomas Sander[7,184], Marcello Scala[9,10], Theresa Scattergood[185], Steven C. Schachter[186], Christoph J. Schankin[142,187], Ingrid E. Scheffer[35,188], Bettina Schmitz[184], Susanne Schoch[33], Susanne Schubert-Bast[135,137], Andreas Schulze-Bonhage[189], Paolo Scudieri[9,10], Pak Sham[28], Beth R. Sheidley[176], Jerry J. Shih[190], Graeme J. Sills[191], Sanjay M. Sisodiya[17,22], Michael C. Smith[192], Philip E. Smith[193], Anja C. M. Sonsma[1], Doug Speed[194,195], Michael R. Sperling[196], Hreinn Stefansson[162], Kári Stefansson[162], Bernhard J. Steinhoff[26,27], Ulrich Stephani[109], William C. Stewart[197,198], Carlotta Stipa[38], Pasquale Striano[9,10], Hans Stroink[199], Adam Strzelczyk[134,135,137], Rainer Surges[30], Toshimitsu Suzuki[200,201], K. Meng Tan[11], R. S. Taneja[6], George A. Tanteles[140], Erik Taubøll[100], Liu Lin Thio[202], G. Neil Thomas[203], Rhys H. Thomas[153,154], Oskari Timonen[123], Paolo Tinuper[38,39], Marian Todaro[11,12], Pınar Topaloğlu[204], Rossana Tozzi[205], Meng-Han Tsai[206], Birute Tumiene[207,208], Dilsad Turkdogan[209], Unnur Unnsteinsdóttir[162], Algirdas Utkus[208], Priya Vaidiswaran[71,72], Luc Valton[210], Andreas van Baalen[109], Annalisa Vetro[23], Eileen P. G. Vining[211], Frank Visscher[212], Sophie von Brauchitsch[135,137], Randi von Wrede[30], Ryan G. Wagner[213], Yvonne G. Weber[44,214], Sarah Weckhuysen[215,216,217], Judith Weisenberg[202], Michael Weller[218], Peter Widdess-Walsh[2,3,81], Markus Wolff[219], Stefan Wolking[214], David Wu[88], Kazuhiro Yamakawa[200,201], Wanling Yang[150], Zuhal Yapıcı[204], Emrah Yücesan[220], Sara Zagaglia[17,22], Felix Zahnert[134], Federico Zara[9,10], Wei Zhou[4,5,60], Fritz Zimprich[141], Gábor Zsurka[30,145] & Quratulain Zulfiqar Ali[14]

[1]Department of Genetics, University Medical Center Utrecht, Utrecht, The Netherlands. [2]School of Pharmacy and Biomolecular Sciences, The Royal College of Surgeons in Ireland, Dublin, Ireland. [3]The FutureNeuro Research Centre, Dublin, Ireland. [4]Analytic and Translational Genetics Unit, Department of Medicine, Massachusetts General Hospital and Harvard Medical School, Boston, MA, USA. [5]Program in Medical and Population Genetics, Broad Institute of MIT and Harvard, Cambridge, MA, USA. [6]Department of Neurology, Vanderbilt University Medical Center, Nashville, TN, USA. [7]Cologne Center for Genomics (CCG), University of Cologne, Faculty of Medicine and University Hospital Cologne, Cologne, Germany. [8]Tel-Aviv University Sackler Faculty of Medicine, Ramat Aviv, Israel. [9]Department of Neurosciences, Rehabilitation, Ophthalmology, Genetics, Maternal and Child Health, University of Genova, Genova, Italy. [10]IRCCS Istituto Giannina Gaslini, Genova, Italy. [11]Department of Medicine, University of Melbourne, Royal Melbourne Hospital, Parkville, Victoria, Australia. [12]Department of Neuroscience, Central Clinical School, Alfred Health, Monash University, Melbourne, Victoria, Australia. [13]Neurology Department, Aneurin Bevan University Health Board, Newport, UK. [14]Adult Genetic Epilepsy Program, University of Toronto, Toronto, Ontario, Canada. [15]Institute for Biomedical Research and Innovation, National Research Council, Cosenza, Italy. [16]St. George's University Hospital NHS Foundation Trust, London, UK. [17]Department of Clinical and Experimental Epilepsy, UCL Queen Square Institute of Neurology, London, UK. [18]Population Health and Immunity Division, The Walter and Eliza Hall Institute of Medical Research, Parkville, Victoria, Australia. [19]Department of Biology, University of Melbourne, Parkville, Victoria, Australia. [20]School of Mathematics and Statistics, University of Melbourne, Parkville, Victoria, Australia. [21]Swansea University Medical School, Swansea University, Swansea, UK. [22]Chalfont Centre for Epilepsy, Chalfont-St-Peter, UK. [23]Pediatric Neurology, Neurogenetics and Neurobiology Unit and Laboratories, Children's Hospital A. Meyer, University of Florence, Florence, Italy. [24]University Health Network, University of Toronto, Toronto, Ontario, Canada. [25]APHM, Timone Hospital, Epileptology and Cerebral Rhythmology, Aix Marseille Univ, INSERM, INS, Inst Neurosci Syst, Marseille, France. [26]Epilepsy Center Kork, Kehl-Kork, Germany. [27]Medical Faculty of the University of Freiburg, Freiburg, Germany. [28]Department of Psychiatry, The University of Hong Kong, Pokulam, Hong Kong. [29]The State Key Laboratory of Brain and Cognitive Sciences, University of Hong Kong, Hong Kong, China. [30]Department of Epileptology, University of Bonn Medical Centre, Bonn, Germany. [31]Department of Neurology, Istanbul Faculty of Medicine, Istanbul University, Istanbul, Turkey. [32]Department of Genetics, Aziz Sancar Institute of Experimental Medicine, Istanbul University, Istanbul, Turkey. [33]Section for Translational Epilepsy Research, Department of Neuropathology, University of Bonn Medical Center, Bonn, Germany. [34]Department of Neurology, University of Ulm, Ulm, Germany. [35]Epilepsy Research Centre, University of Melbourne, Austin Health, Heidelberg, Victoria, Australia. [36]Stichting Epilepsie Instellingen Nederland (SEIN), Zwolle, The Netherlands. [37]Department of Neurology, American University of Beirut Medical Center, Beirut, Lebanon. [38]IRCCS Istituto delle Scienze Neurologiche di Bologna, Bologna, Italy. [39]Department of Biomedical and Neuromotor Sciences, University of Bologna, Bologna, Italy. [40]Department of Neurology, Sheba Medical Center, Ramat Gan, Israel. [41]Luxembourg Centre for Systems Biomedicine, University of Luxembourg, Esch-sur-Alzette, Luxembourg. [42]Department of Pediatric Neurology, Dr von Hauner Children's Hospital, Ludwig Maximilians University, Munich, Germany. [43]Epilepsy Center Munich, Munich, Germany. [44]Department of Neurology and Epileptology, Hertie Institute for Clinical Brain Research, University of Tübingen, Tübingen, Germany. [45]Center for Applied Genomics, The Children's Hospital of Philadelphia, Philadelphia, PA, USA. [46]Quantinuum Research LLC, Wayne, PA, USA. [47]Children's Hospital, Department of Pediatric Neurology, University Medical Center Göttingen, Göttingen, Germany. [48]National Human Genome Research Institute, National Institutes of Health, Bethesda, MD, USA. [49]Department of Biomedical Sciences, Cooper Medical School of Rowan University, Camden, NJ, USA. [50]Department of Neurology, Thomas Jefferson University Hospital, Philadelphia, PA, USA. [51]Genomic Medicine Institute, Lerner Research Institute, Cleveland Clinic, Cleveland, OH, USA. [52]Cleveland Clinic Epilepsy Center, Neurological Institute, Cleveland Clinic, Cleveland, OH, USA. [53]Department of Neurology, Neurological Institute, Cleveland Clinic, Cleveland, OH, USA. [54]Department of Molecular Biology and Genetics, Bogaziçi University, Istanbul, Turkey. [55]Belfast Health and Social Care Trust, Belfast, UK. [56]Integrated Diagnostics for Epilepsy, Fondazione IRCCS Istituto Neurologico C. Besta, Milan, Italy. [57]Hereditary Research Lab, Bethlehem University, Bethlehem, Palestine. [58]Division of Epilepsy, Department of Neurology, Mayo Clinic, Rochester, MN, USA. [59]Unit of Genetics of Neurodegenerative and Metabolic Diseases, Fondazione IRCCS Istituto Neurologico Carlo Besta, Milan, Italy. [60]Stanley Center for Psychiatric Research, Broad Institute of Harvard and M.I.T, Cambridge, MA, USA. [61]Hôpital Lariboisière, Dept of Neurosurgery-Paris-Cité University, Paris, France. [62]Department of Epidemiology and Preventive Medicine, School of Public Health, Sackler Faculty of Medicine, Tel Aviv University, Tel Aviv, Israel. [63]Department of Pharmacology and Pharmacy, The University of Hong Kong, Pokfulam, Hong Kong. [64]Department of Pediatric Neurology, Chang Gung Memorial Hospital, Linkou Branch, and College of Medicine, Chang Gung University, Taoyuan, Taiwan. [65]Kids Research, Children's Hospital at Westmead Clinical School, Faculty of Medicine and Health, University of Sydney, Sydney, New South Wales, Australia. [66]Neurology Research Group, Swansea University Medical School, Faculty of Medicine, Health & Life Science, Swansea University, Swansea, UK. [67]Cincinnati Children's Hospital Medical Center, Cincinnati, OH, USA. [68]Neurology, Massachusetts General Hospital, Boston, MA, USA. [69]Department of Clinical Neurosciences, Cambridge Biomedical Campus, Cambridge, UK. [70]Department of Neuroscience, Reproductive and Odontostomatological Sciences, University Federico II, Naples, Italy. [71]Division of Neurology, Children's Hospital of Philadelphia, Philadelphia, PA, USA. [72]The Epilepsy NeuroGenetics Initiative (ENGIN), Children's Hospital of Philadelphia, Philadelphia, PA, USA. [73]Department of Neurosciences, Université de Montréal, Montréal, Quebec, Canada. [74]Department of Neurology, Royal Victoria Hospital, Belfast Health and Social Care Trust, Belfast, UK. [75]Institute for Molecular Medicine Finland (FIMM), University of Helsinki, Helsinki, Finland. [76]Division of Genetic Medicine, Department of Medicine, Vanderbilt University Medical Center, Nashville, TN, USA. [77]Department of Psychiatry and Behavioral Sciences, Vanderbilt University Medical Center, Nashville, TN, USA. [78]Department of Biomedical Informatics, Vanderbilt University Medical Center, Nashville, TN, USA. [79]Vanderbilt Genetics Institute, Vanderbilt University Medical Center, Nashville, TN, USA. [80]Stichting Epilepsie Instellingen Nederland (SEIN), Heemstede, The Netherlands. [81]Department of Neurology, Beaumont Hospital, Dublin, Ireland. [82]Department of Neurology, Hôpital Erasme, Université Libre de Bruxelles, Bruxelles, Belgium. [83]Department of Clinical Neurophysiology, Lille University Medical Center, University of Lille, Lille, France. [84]Department of Neurology, New York University/Langone Health, New York City, NY, USA. [85]Department of Neurology, St. James's Hospital, Dublin, Ireland. [86]Department of Neurology, University of Pennsylvania, Perelman School of Medicine, Philadelphia, PA, USA. [87]Department of General Practice and Primary Health Care, University of Helsinki and Helsinki University Hospital, Helsinki, Finland. [88]Human Genetics Training Program, Vanderbilt University, Nashville, TN, USA. [89]Psychiatric & Neurodevelopmental Genetics Unit, Department of Psychiatry, Massachusetts General Hospital and Harvard Medical School, Boston, MA, USA. [90]Division of Biostatistics, Institute of Epidemiology and Preventive Medicine, College of Public Health, National Taiwan University, Taipei, Taiwan. [91]Department of Pharmacology and Psychiatry, University of Pennsylvania Perlman School of Medicine, Philadelphia, PA, USA. [92]Department of Pediatrics and Neonatology, Medical University of Vienna, Vienna, Austria. [93]Institute of Neurology, Department of Medical and Surgical Sciences, University 'Magna Graecia', Catanzaro, Italy. [94]Neurophysiology, Fondazione IRCCS Istituto Neurologico Carlo Besta, Milan, Italy. [95]Institute of Clinical Molecular Biology, Christian-Albrechts-University of Kiel, University Hospital Schleswig Holstein, Kiel, Germany. [96]Department of Neurology, NYU School of Medicine, New York City, NY, USA. [97]Department of Pediatric Neuroscience, Fondazione IRCCS Istituto Neurologico Carlo Besta, Milan, Italy. [98]Department of Medical and Surgical Sciences, Neuroscience Research Center, Magna Graecia University, Catanzaro, Italy. [99]Institute of Neurology and Neurosurgery at St. Barnabas, Livingston, NJ, USA. [100]Department of Neurology, Division of Clinical Neuroscience, Rikshospitalet Medical Centre, University of Oslo, Oslo, Norway.

[101]Department of Neurology, Baylor College of Medicine, Houston, TX, USA. [102]Department of Pediatrics, Nationwide Children's Hospital, Columbia, OH, USA. [103]Division of Human Genetics, Department of Pediatrics, The Perelman School of Medicine, University of Pennsylvania, Philadelphia, PA, USA. [104]Life and Brain Center, University of Bonn Medical Center, Bonn, Germany. [105]Institute for Genomic Statistics and Bioinformatics, University of Bonn, Bonn, Germany. [106]Department of Neurology, University of California, San Francisco, CA, USA. [107]Division of Pharmacotherapy and Experimental Therapeutics, Eshelman School of Pharmacy, University of North Carolina at Chapel Hill, Chapel Hill, NC, USA. [108]Department of Genetics, School of Medicine, University of North Carolina at Chapel Hill, Chapel Hill, NC, USA. [109]Department of Neuropediatrics, University Medical Center Schleswig-Holstein, Christian-Albrechts-University, Kiel, Germany. [110]Department of Biomedical and Health Informatics (DBHi), Children's Hospital of Philadelphia, Philadelphia, PA, USA. [111]Hasso Plattner Institute, Digital Health Center, University of Potsdam, Potsdam, Germany. [112]Hasso Plattner Institute, Mount Sinai School of Medicine, New York City, NY, USA. [113]General Medical Research Center, School of Medicine, Fukuoka University, Fukuoka, Japan. [114]Department of Neurology, University Hospital of Strasbourg, Strasbourg, France. [115]Danish Epilepsy Centre, Dianalund, Denmark. [116]Institute of Regional Health Services Research, University of Southern Denmark, Odense, Denmark. [117]Swiss Epilepsy Center, Klinik Lengg, Zurich, Switzerland. [118]National Epilepsy Center, Shizuoka Institute of Epilepsy and Neurological Disorder, Shizuoka, Japan. [119]Department of Pediatrics, Fukuoka Sanno Hospital, Fukuoka, Japan. [120]Department of Psychiatry and Applied Psychology, Institute of Mental Health University of Nottingham, Nottingham, UK. [121]Division of Brain Sciences, Imperial College London, London, UK. [122]Kuopio Epilepsy Center, Neurocenter, Kuopio University Hospital, Kuopio, Finland. [123]Institute of Clinical Medicine, University of Eastern Finland, Kuopio, Finland. [124]Department of Computational Biology and Medical Sciences, Graduate School of Frontier Sciences, The University of Tokyo, Tokyo, Japan. [125]The Broad Institute of M.I.T. and Harvard, Cambridge, MA, USA. [126]Department of Statistical Genetics, Osaka University Graduate School of Medicine, Suita, Japan. [127]Department of Child Neurology, Medical School, Kocaeli University, Kocaeli, Turkey. [128]Neuroscience Unit, KEMRI-Wellcome Trust Research Programme, Kilifi, Kenya. [129]Department of Public Health, Pwani University, Kilifi, Kenya. [130]Department of Psychiatry, University of Oxford, Oxford, UK. [131]Department of Pediatrics, Showa University School of Medicine, Epilepsy Medical Center, Showa University Hospital, Tokyo, Japan. [132]Department of Pharmacology and Toxicology, American University of Beirut Faculty of Medicine, Beirut, Lebanon. [133]Department of Paediatrics and Child Health, University of Otago, Wellington, New Zealand. [134]Epilepsy Center Hessen-Marburg, Department of Neurology, Philipps University Marburg, Marburg, Germany. [135]Epilepsy Center Frankfurt Rhine-Main, Center of Neurology and Neurosurgery, Goethe University Frankfurt, Frankfurt, Germany. [136]Departments of Clinical Neurosciences, Medical Genetics and Community Health Sciences, Hotchkiss Brain Institute & Alberta Children's Hospital Research Institute, Cumming School of Medicine, University of Calgary, Calgary, Alberta, Canada. [137]LOEWE Center for Personalized Translational Epilepsy Research (CePTER), Goethe University Frankfurt, Frankfurt, Germany. [138]Neuropediatric Clinic and Clinic for Neurorehabilitation, Epilepsy Center for Children and Adolescents, Vogtareuth, Germany. [139]Research Institute for Rehabilitation, Transition, and Palliation, Paracelsus Medical University, Salzburg, Austria. [140]Cyprus Institute of Neurology and Genetics, Nicosia, Cyprus. [141]Department of Neurology, Medical University of Vienna, Vienna, Austria. [142]Department of Neurology, Inselspital, Bern University Hospital, University of Bern, Bern, Switzerland. [143]Yale School of Medicine, New Haven, CT, USA. [144]Institute of Human Genetics, University of Leipzig Medical Center, Leipzig, Germany. [145]Institute of Experimental Epileptology and Cognition Research, Medical Faculty, University of Bonn, Bonn, Germany. [146]Neuropediatrics Department, Bonifatius Hospital Lingen, Lingen, Germany. [147]Department of Neurology, Hofstra-Northwell Medical School, New York City, NY, USA. [148]Department of Medicine and Therapeutics, Chinese University of Hong Kong, Hong Kong, China. [149]Department of Biomedical and Dental Sciences, Morphological and Functional Images (BIOMORF), University of Messina, Messina, Italy. [150]Department of Paediatrics and Adolescent Medicine, The University of Hong Kong, Hong Kong, Hong Kong. [151]Folkhälsan Research Center and Medical Faculty, University of Helsinki, Helsinki, Finland. [152]Department of Medical Genetics, Hospices Civils de Lyon and University of Lyon, Lyon, France. [153]Translational and Clinical Research Institute, Newcastle University, Newcastle Upon Tyne, UK. [154]Department of Clinical Neurosciences, Newcastle Upon Tyne Hospitals NHS Foundation Trust, Newcastle Upon Tyne, UK. [155]Department of Health Technology and Informatics, The Hong Kong Polytechnic University, Hung Hum, Hong Kong. [156]Neuroscience Department, Janssen Research & Development, LLC, Titusville, NJ, USA. [157]Child Neurology, New Children's Hospital, Helsinki, Finland. [158]Pediatric Research Center, University of Helsinki, Helsinki, Finland. [159]Helsinki University Hospital, Helsinki, Finland. [160]Department of Translational Medicine, School of Medical Sciences, University of Campinas (UNICAMP), and the Brazilian Institute of Neuroscience and Neurotecnology, Campinas, Brazil. [161]Department of Medicine, Tseung Kwan O Hospital, Tseung Kwan O, Hong Kong. [162]deCODE genetics, Reykjavík, Iceland. [163]Department of Pharmacology and Therapeutics, University of Liverpool, Liverpool, UK. [164]Division of Intramural Population Health Research, Eunice Kennedy Shriver National Institute of Child Health and Human Development, National Institutes of Health, Bethesda, MD, USA. [165]School of Medicine, Trinity College Dublin, Dublin, Ireland. [166]Epilepsy Center for Children, University Hospital Ruppin-Brandenburg, Brandenburg Medical School, Neuruppin, Germany. [167]Pediatric Neurology, University of Giessen, Giessen, Germany. [168]Institute of Human Genetics, University of Bonn Medical Center, Bonn, Germany. [169]University Hospital Cologne, Cologne, Germany. [170]Laboratory for Systems Genetics, RIKEN Center for Integrative Medical Sciences, Yokohama, Japan. [171]Department of Neurology, Landspitalinn University Hospital, Reykjavik, Iceland. [172]Istanbul University-Cerrahpaşa, Cerrahpaşa Medical Faculty, Department of Neurology, Istanbul, Turkey. [173]Department of Psychiatry, Robert Wood Johnson Medical School and New Jersey Medical School, Rutgers University, Newark, NJ, USA. [174]Centre for Genomics Research, Discovery Sciences, BioPharmaceuticals R&D, AstraZeneca, Cambridge, UK. [175]Department of Neurology, Morriston Hospital, Swansea Bay University Bay Health Board, Swansea, UK. [176]Epilepsy Genetics Program, Division of Epilepsy and Clinical Neurophysiology, Department of Neurology, Boston Children's Hospital, Boston, MA, USA. [177]IRCCS Azienda Ospedaliero-Universitaria di Bologna, Medical Genetics Unit, Bologna, Italy. [178]Department of Neurology, Gardner Neuroscience Institute, University of Cincinnati Medical Center, Cincinnati, OH, USA. [179]Department of Neurology, Duke University School of Medicine, Durham, NC, USA. [180]Faculty of Medicine & Health, University of Sydney, Sydney, New South Wales, Australia. [181]Department of Functional Neurology and Epileptology, Hospices Civils de Lyon and University of Lyon, Lyon, France. [182]Lyon Neuroscience Research Center, INSERM, Lyon, France. [183]Department of Clinical Neurosciences, Centre Hospitalo-Universitaire Vaudois, Lausanne, Switzerland. [184]Department of Neurology, Charité Universitaetsmedizin Berlin, Campus Virchow-Clinic, Berlin, Germany. [185]Department of Endocrinology, Hospital of The University of Pennsylvania, Philadelphia, PA, USA. [186]Departments of Neurology, Beth Israel Deaconess Medical Center, Massachusetts General Hospital, and Harvard Medical School, Boston, MA, USA. [187]Department of Neurology, Ludwig Maximilians University, Munchen, Germany. [188]Department of Neurology, Royal Children's Hospital, Parkville, Victoria, Australia. [189]Department of Epileptology, University Hospital Freiburg, Freiburg, Germany. [190]Department of Neurosciences, University of California, San Diego, CA, USA. [191]School of Life Sciences, University of Glasgow, Glasgow, UK. [192]Rush University Medical Center, Chicago, IL, USA. [193]Department of Neurology, Alan Richens Epilepsy Unit, University Hospital of Wales, Cardiff, UK. [194]UCL Genetics Institute, University College London, London, UK. [195]Aarhus Institute of Advanced Studies (AIAS), Aarhus University, Aarhus, Denmark. [196]Department of Neurology and Comprehensive Epilepsy Center, Thomas Jefferson University, Philadelphia, PA, USA. [197]Department of Pediatrics, Ohio State University, Columbus, OH, USA. [198]The Research Institute, Nationwide Children's Hospital, Columbus, OH, USA. [199]CWZ Hospital, Nijmegen, The Netherlands. [200]Department of Neurodevelopmental Disorder Genetics, Institute of Brain Science, Nagoya City University Graduate School of Medical Science, Nagoya, Japan.

[201]Laboratory for Neurogenetics, RIKEN Center for Brain Science, Wako, Japan. [202]Department of Neurology, Washington University School of Medicine, St. Louis, MO, USA. [203]Institute for Applied Health Research,  University of Birmingham, Birmingham, UK. [204]Department of Child Neurology, Istanbul Faculty of Medicine, Istanbul University, Istanbul, Turkey. [205]C. Mondino National Neurological Institute, Pavia, Italy. [206]Department of Neurology, Kaohsiung Chang Gung Memorial Hospital, Kaohsiung, Taiwan. [207]Centre for Medical Genetics, Vilnius University Hospital Santaros Klinikos, Vilnius, Lithuania. [208]Institute of Biomedical Sciences, Faculty of Medicine, Vilnius University, Vilnius, Lithuania. [209]Department of Child Neurology, Medical School, Marmara University, Istanbul, Turkey. [210]Epilepsy Unit, Department of Neurology, Brain and Cognition Research Center, University Hospital and University of Toulouse, Paul Sabatier University, Toulouse, France. [211]Departments of Neurology and Pediatrics, The Johns Hopkins University School of Medicine, Baltimore, MD, USA. [212]Department of Neurology, Admiraal De Ruyter Hospital, Goes, The Netherlands. [213]MRC/Wits Rural Public Health & Health Transitions Research Unit (Agincourt), School of Public Health, Faculty of Health Sciences, University of the Witwatersrand, Johannesburg, South Africa. [214]Department of Neurology and Epileptology, University of Aachen, Aachen, Germany. [215]Applied & Translational Neurogenomics Group, VIB Center for Molecular Neurology, VIB, Antwerp, Belgium. [216]Department of Neurology, Antwerp University Hospital, Edegem, Belgium. [217]Translational Neurosciences, Faculty of Medicine and Health Science, University of Antwerp, Antwerp, Belgium. [218]Department of Neurology, University Hospital and University of Zurich, Zürich, Switzerland. [219]Department of Pediatric Neurology, Vivantes Hospital Neukölln, Berlin, Germany. [220]Bezmialem Vakif University, Institute of Life Sciences and Biotechnology, Istanbul, Turkey. ✉e-mail: s.berkovic@unimelb.edu.au; gcavalleri@rcsi.ie; b.p.c.koeleman@umcutrecht.nl

## Methods

### Inclusion and ethics statement

Local institutional review boards approved study protocols at each contributing site. All study participants provided written, informed consent for the use of their data in genetic studies of epilepsy. For minors, written informed consent was obtained from their parents or legal guardian.

### Sample and phenotype descriptions

This meta-analysis combines previously published datasets with new genotyped cohorts. Descriptions of the 24 cohorts included in our previous analysis can be found in the Supplementary Table 6 of that publication[4]. Here we included five new cohorts (Supplementary Table 1), comprising 14,732 epilepsy cases and 22,362 controls, resulting in a total sample size of 29,944 cases and 52,538 controls. Classification of epilepsy was performed as described previously (see Supplementary Note for a detailed description)[4]. In brief, we assigned people with epilepsy to FE, GGE or unclassified epilepsy. 'All epilepsy' was the combination of GGE, focal and unclassified epilepsy. Where possible, we used EEG, MRI and clinical history to further refine the subphenotypes—JME, CAE, JAE, GTCSA, nonlesional FE, FE with HS and FE with lesions other than HS.

### Genotyping, quality control (QC) and imputation

Study participants were genotyped on SNP arrays (see Supplementary Table 1 for an overview of genotyping in new cohorts). QC was performed separately for each cohort. Pre-imputation QC included removal of SNPs with call rate (<98%), differential missing rate, duplicated and monomorphic SNPs, SNPs with batch association ($P < 10^{-4}$) and violation of Hardy–Weinberg equilibrium ($P < 10^{-10}$). In addition, the Epi25 cohort was split by ancestry, based on principal component analysis. Individuals were removed if their heterozygous/homozygous ratio was >4 s.d. from the mean. We also removed one from each pair of related samples (determined by identity-by-descent >0.2) and removed individuals with ambiguous or nonmatching genetically imputed sex. Furthermore, 3,180 duplicates between the Epi25 cohort and the previously published genome-wide mega-analysis[4] were identified based on genotype and were removed from the Epi25 cohort. Of the 3,180 duplicates, 1,226 were GGE and 1,402 FE. Before imputation, cohorts were cross-referenced to the Haplotype Reference Consortium (HRC) panel to ensure SNPs matched in terms of strand, position and ref/alt allele assignment. Additionally, SNPs were removed if they were absent in the HRC panel, if they had a >20% allele frequency difference with the HRC panel or if any AT/GC SNPs had MAFs >40%, using tools available from https://www.well.ox.ac.uk/~wrayner/tools/. Data from Janssen Pharmaceuticals, Austrian GenEpa, Swiss GenEpa, Norwegian GenEpa and BPCCC were then imputed using the Wellcome Sanger Institutes' imputation server (https://imputation.sanger.ac.uk/), using EAGLE v2.4.1 (ref. 60) for phasing, and the Positional Burrows–Wheeler Transform algorithm[61] v3.1 for imputation. The HRC reference panel r1.1 was used as a reference for imputation ($n = 32,470$) (ref. 62). Similarly, data from the Epi25 cohort were imputed using the Michigan Imputation server (https://imputationserver.sph.umich.edu/). We used the HRC r1.1 as the reference panel for individuals of European and Asian ancestry and the 1000 Genomes Phase 3 v5 ($n = 2,504$) for individuals of African ancestry. Default imputation parameters were used. Due to data sharing restrictions and with the Epi25 cohort data located in the USA and the other cohorts located in the European Union, we were unable to merge the data or use the same imputation server. Postimputation QC was largely similar among all cohorts. The Epi25 cohort used an in-house pipeline, where imputed dosages were used for genome-wide association analyses, filtering on imputation INFO > 0.3, MAF < 1%, genotype coverage <0.98 and Hardy–Weinberg violations ($P < 10^{-5}$). For all other cohorts, the same procedures as our previous study[4] were used—imputed datasets were

converted to hard-coded PLINK format, requiring a more stringent imputation filtering of INFO > 0.9 (as opposed to dosages, where imputation inaccuracy is incorporated in downstream analyses). Furthermore, we removed SNPs with MAF < 5%, genotype coverage <0.98 and Hardy–Weinberg violations ($P < 10^{-5}$) (ref. 4). We removed SNPs <5% MAF in the Janssen Pharmaceuticals, Austrian GenEpa, Swiss GenEpa, Norwegian GenEpa and BPCCC cohorts for QC reasons, and note there will be a corresponding loss in study power for lower frequency SNPs in the 'focal' and 'all epilepsy' epilepsy analysis.

### Genome-wide association analyses

GWAS of the Janssen Pharmaceuticals, Swiss GenEpa, Norwegian GenEpa and Austrian GenEpa cohorts was performed as a mega-analysis, as described previously[4]. GWAS of the Epi25 cohort was performed with a generalized mixed model using SAIGE v0.38 (ref. 63). SAIGE was performed in two steps. First, we fit the null logistic mixed model to estimate the variance component and other model parameters. For this step, SNPs were filtered on-call rate >0.98 and MAF > 5%, and SNPs were pruned to obtain approximate independent markers (window size of 100 SNPs and $r^2 > 0.3$). Second, we tested for the association between each genetic variant and phenotypes by applying SPA to the score test statistics. Next, we performed $P$ value-based fixed-effects meta-analyses with METAL v2020-05-05 (ref. 64) for each of the main phenotypes ('all', GGE and FE), as well as the subphenotypes, weighted by effective samples sizes ($n_{eff} = 4/(1/n_{cases} + 1/n_{controls})$) to account for case–control imbalance. We performed multi-ancestry and European-only meta-analyses for the main phenotypes, and restricted the subphenotype analyses to Europeans only, due to limited sample size in other ancestries. We included all SNPs (~4.9 million, MAF > 1%) that were present in at least the previous mega-analysis and the Epi25 dataset, which together account for 88% of the total sample size. We calculated genomic inflation factors ($\lambda$), mean $\chi^2$ and LD-score regression intercepts to assess potential inflation of the test statistic. Because $\lambda$ is known to scale with sample size, we also calculated $\lambda1000$, which is $\lambda$ corrected for an equivalent sample size of 1,000 cases and 1,000 controls[65]. We limited these analyses to participants of European ancestry because LD-structure depends on ethnicity and Europeans constituted 92% of cases. For forest plots of genome-wide significant hits, Beta/SE was estimated from METAL $z$ scores using a previously published formula[22]. For P–M plots, $m$ values were generated using the default settings of the tool Metasoft v2.0.0 (ref. 66).

### Data sources for the biobank and deCODE genetics GWAS

Summary statistics for epilepsy GWAS were obtained from three population biobanks (UK Biobank[67], Biobank Japan[68,69] and FinnGen release R6 (ref. 70)) and from deCODE genetics[71] (Iceland). The Biobank Japan, FinnGen and deCODE genetics epilepsy cases were further assigned into either 'focal' or 'generalized' epilepsy, whereas the UK Biobank samples were not subdivided based on seizure localization, as the relevant clinical details were unavailable to facilitate an accurate subdivision (see Supplementary Table 14 for sample sizes per biobank and deCODE genetics). Control data were population-matched samples with no history of epilepsy.

Fixed-effects meta-analyses were conducted using METAL v2020-05-05 (ref. 64), weighted by effective sample size ($n_{eff} = 4/(1/n_{cases} + 1/n_{controls})$) to account for case–control imbalance.

**UK Biobank.** We identified people with epilepsy from the UK Biobank using an analysis of self-reported data, inpatient hospital episode statistics, death certificate diagnostic data and primary care diagnostic data as described elsewhere[72]. This allowed us to interrogate the evidence available to support a diagnosis of epilepsy rather than relying purely on UK Biobank-generated data fields 131048 and 13049 based on ICD-10 G40 mapping.

**FinnGen.** Epilepsy was determined with ICD-10 G40, ICD-9 345, ICD-8 345 and Social Insurance Institution of Finland (KELA) code 111. Exclusion criteria were ICD-9 3452/3453 and ICD-8 34520. GGE was determined with ICD-10 G40.3, ICD-9 345(0-3) and ICD-8 34519. Exclusion criteria were ICD-8 34511. FE was determined with ICD-10 G40.0, G40.1, G40.2, ICD-9 345(45) and ICD-8 3453.

**deCODE genetics.** Epilepsy was determined with ICD-10 G40 and ICD-9 345 excluding 3452/3453. GGE with ICD-10 G40.3/G40.4/G40.6/G40.7 or ICD-9 3450/3451/3456, and FE with ICD-10 G40.0/G40.1/G40.2 or ICD-9 3454/3455.

**Biobank Japan.** Cases were classified into 'Broad_Epilepsy,' being any form of epilepsy; 'Idiopathic_Epilepsy,' being epilepsy with onset under 40 years and no known cause or 'Idiopathic_Focal_Epilepsy' and 'Idiopathic_Generalized_Epilepsy,' where focal and generalized syndromes could be ascertained.

Control data were population-matched samples with no history of epilepsy. GWAS fixed-effects meta-analyses were conducted using METAL[64]. To account for case–control imbalance, the effective sample size for each cohort was calculated as $n_{eff} = 4/(1/n_{cases} + 1/n_{controls})$. GWAS Manhattan plots were generated using the qqman package[73] in R v3.6.0. Genome-wide significant loci were mapped onto genes using the FUMA web platform[18].

We performed three meta-analyses. As a primary analysis, we meta-analyzed all nonbiobank samples, then we meta-analyzed only biobank/deCODE genetics samples and finally, we performed a combined meta-analysis of biobank/deCODE genetics and nonbiobank samples.

### Pleiotropy analysis
ASSET[74] is a meta-analysis-based pleiotropy detection approach that identifies common or shared genetic effects between two or more related, but distinct traits. We used ASSET v2.2.0 with a genome-wide significance level of $\alpha = 5 \times 10^{-8}$. We applied ASSET to the subset of European-ancestry samples, comprising 6,952 (3,244 + 3,708) GGE cases and 14,939 (5,344 + 9,095) FE cases from the Epi25 and our consortium as well as 42,434 partially overlapping controls from both consortia. Note that ASSET accounts for sample overlap in the analysis. Effect sizes, standard errors and the effective sample sizes estimated were from the main meta-analysis.

### HLA association
Given the prior association of the HLA with autoimmune epilepsy[75,76], we included a specific analysis of the HLA. HLA types and amino acid residues were imputed using CookHLA software v1.0.1 (ref. 26), with the 1000 Genomes Phase 3 used as a reference panel[77]. Samples were grouped by genetic ancestry for imputation.

Following imputation, association analysis was conducted using the HLA Analysis Toolkit (HATK) v1.2 (ref. 78). The following three phenotypes were analyzed: 'all epilepsy', FE and GGE. Samples from the ILAE and Epi25 datasets were analyzed separately, and the association results were meta-analyzed across datasets and ancestries using PLINK v1.9 (ref. 79).

### Functional annotation
We annotated all genome-wide significant SNPs and tagged SNPs within the loci from our multi-ancestry meta-analyses. ANNOVAR v2017-07-17 was used to retrieve the location and function of each SNP[80], the CADD score was used as a measure of predicted deleteriousness[81] and chromatin states were incorporated from the ENCODE and NIH Roadmap Epigenomics Mapping Consortium[14,82]. We used FUMA v1.3.8 to define the independently significant SNPs within loci; that is, SNPs that were genome-wide significant but not in LD ($r^2 < 0.2$ in Europeans) with the lead SNP in the locus.

### MTAG
MTAG v1.0.8 (ref. 17) was used (with default settings) to increase the effective sample size from our European ancestry GGE subphenotype analysis by pairing it with the strongly correlated overall GGE GWAS with a larger sample size. MTAG accounts for sample overlap between traits and uses the fact that estimations of effect size and standard error of a primary GWAS, in this case GGE subtypes, can be improved by matching them to a genetically correlated secondary GWAS, in this case GGE[17]. Similarly, we applied MTAG to combine FE with GGE.

### Gene mapping
To map genome-wide significant loci from our multi-ancestry meta-analyses to specific genes, we used FUMA v1.3.8 (ref. 18) with the same parameters as published previously[4]. We defined genome-wide significant loci as the region encompassing all SNPs with $P < 10^{-4}$ that were in LD ($r^2 > 0.2$) with the lead SNP (that is, the SNP with the strongest association within the region). We used a combination of positional mapping (within 250 kb from the locus), eQTL mapping (SNPs with FDR corrected eQTL $P < 0.05$ in blood or brain tissue) and 3D Chromatin Interaction Mapping (FDR $P < 10^{-6}$ in brain tissue).

### Genome-wide gene-based association study (GWGAS) and gene-set analyses
We performed the GWGAS using the default settings of MAGMA v1.08, as implemented in FUMA v1.3.8, which calculates an association $P$ value based on all the associations of all SNPs within each gene in the GWAS[19]. Based on these GWGAS results, we performed competitive gene-set analyses with default MAGMA settings, using 15,483 default gene sets and GO-terms from MsigDB. In addition, we specifically assessed 18 curated gene sets involving different synaptic functions[30].

### TWAS
TWAS was performed with FUSION v3, with default settings[20]. We imputed gene expression based on our European-only GWAS (because the method relies on LD reference data) eQTL data from the PsychENCODE consortium, which includes dorsolateral prefrontal cortex tissue from 1,695 individuals[21].

### SMR
SMR v1.03 is an additional method to assess the association between epilepsy and expression of specific genes[22]. Although TWAS and SMR have similar aims, the differences in methods and reference datasets result in complementary information. As opposed to the FUSION TWAS method, which uses multi-SNP imputation of gene expression, SMR uses Mendelian randomization to test whether the effect size of an SNP on epilepsy is mediated by the expression of specific genes. We performed SMR analyses with default settings, using European-only GWAS and the MetaBrain expression data as reference, a new eQTL dataset including 2,970 human brain samples[83].

### Sex-specific analyses
We performed a GWAS, as described above, for all epilepsy (13,889 female cases and 19,676 female controls; 12,259 male cases and 18,645 male controls) and GGE (3,946 female cases and 19,676 female controls; 2,603 male cases and 18,645 male controls) separately for participants of either sex, after which we performed fixed-effects meta-analyses with METAL to merge the different cohorts. We performed meta-analyses between the male and female GWAS with GWAMA v2.2.2 (ref. 84) to assess the heterogeneity of effect sizes between sexes and sex-differentiated associations[35]. Sex-differentiated analyses are meta-analyses between female-only and male-only GWAS, allowing for different effect sizes between the sexes, while sex-heterogeneity tests the difference in effect size for each SNP between female-only and male-only GWAS[35].

## Gene prioritization

We combined ten methods to prioritize the most likely biological candidate gene within each genome-wide significant locus. For each gene in each locus, we assessed the following criteria:

- Missense: we assessed whether the SNPs tagged in the genome-wide significant locus contained an exonic missense variant in the gene, as annotated by ANNOVAR v2017-07-17.
- TWAS: we assessed whether imputed gene expression was significantly associated with the epilepsy phenotype, based on the FUSION TWAS as described above, Bonferroni corrected for each mapped gene with expression information.
- SMR: we assessed whether the gene had a significant SMR association with the epilepsy phenotype, based on the SMR analyses as described above, Bonferroni corrected for each mapped gene with expression information.
- MAGMA: we assessed whether the gene was significantly associated with the epilepsy phenotype through a GWGAS analysis, Bonferroni corrected for each mapped gene.
- PoPS: we calculated the polygenic priority score (PoPS)[85], a method that combines GWAS summary statistics with biological pathways, gene expression and protein–protein interaction data, to pinpoint the most likely causal genes. We scored the gene with the highest PoPS score within each locus.
- Brain expression: for each mapped gene, we calculated the mean expression in all brain and nonbrain tissues based on data from the GTEx project v8 (ref. [86]). Next, we assessed whether the gene was more strongly expressed in brain tissues than nonbrain tissues, by comparing the average expression in all brain tissues with all nonbrain tissues.
- Brain-coX: we assessed whether genes were prioritized as co-expressed with established epilepsy genes in more than a third of brain tissue resources used, using the tool brain-coX (Supplementary Fig. 26)[87].
- Target of AED: we assessed whether the gene is a known target of an anti-epileptic drug, as detailed in the drug–gene interaction database (www.DGidb.com; accessed on 26-11-2021) and a list of drug targets from a recent publication (Supplementary Data 10)[88].
- Knockout mouse: we assessed whether a knockout of the gene in a mouse model results in a nervous system (phenotype ID: MP:0003631) or a neurological/behavior phenotype (MP:0005386) in the Mouse Genome Informatics database (http://www.informatics.jax.org; accessed on 26-11-2021).
- Monogenic epilepsy gene: we evaluated whether the gene is listed as a monogenic epilepsy gene, in a curated list maintained by the Epilepsy Research Center at the University of Melbourne[89] (Supplementary Data 10).

Similar to previous studies[4,90], we scored all genes based on the number of criteria being met (range: 0–10; all criteria had an equal weight). The gene with the highest score was chosen as the most likely implicated (see Supplementary Data 6 for a complete list of scores for all genes in each locus). We implicated both genes if they had an identical, highest score. We calculated Pearson correlation coefficients between the ten criteria (Supplementary Table 16) and note that most correlations were low (range: −0.13 to 0.39), suggesting that they convey complementary information.

## Long-distance expression regulation of BCL11A

Most eQTL databases, like PsychENCODE and MetaBrain, restrict eQTL analyses to 1 Mb distance between genes and SNPs. To specifically assess the hypothesis of long-distance regulation of *BCL11A* by the lead SNPs in the 2p16.1 epilepsy locus, we manually interrogated the MetaBrain database[83] without distance restraints. Next, we calculated the association between the three lead SNPs in the locus (rs11688767, rs77876353 and rs13416557) with *BCL11A* expression.

## Heritability analyses

We calculated SNP-based heritability on the European-only GWAS using LDAK v5.2, as it was recently shown to give more accurate heritability estimates for complex traits, when compared to other methods including LDSC[91,92]. We used default settings in LDAK and precalculated LD weights from 2,000 European (white British) reference samples under the BLD–LDAK SumHer model[92]. SNP-based heritabilities were converted to liability scale heritability estimates, using the following formula: $h^2_1 = h^2_o \times K^2(1 - K)^2/p(1 - p) \times Z^2$, where $K$ is the disease prevalence, $p$ is the proportion of cases in the sample and $Z$ is the standard normal density at the liability threshold. To decrease downward bias, we performed these calculations based on the effective sample sizes (see calculation above), after which $p = 0.5$ can be assumed[93], with the same population prevalences as our previous study (Supplementary Table 10)[4]. The total amount of causally associated variants (that is, variants with nonzero additive genetic effect) underlying epilepsy risk was calculated by a causal mixture model (MiXeR) v1.2.0 (ref. [38]). MiXeR uses a likelihood-based framework to estimate the amount of causal SNPs underlying a trait, without the need to pinpoint which specific SNPs are involved. Furthermore, MiXeR allows for power calculations to assess the required sample size to explain a certain proportion of SNP-based heritability by genome-wide significant SNPs.

## Genomic SEM

Genomic SEM entails two stages of estimation[29]. In the first stage, the empirical genetic covariance matrix and sampling covariance matrix are estimated using an extension of multivariable LDSC. This matrix is extended to include SNP effects for the multivariate GWAS SEM. In the second stage, an SEM is specified, and its parameters are estimated such that the discrepancies in the model covariance matrix and the empirical covariance matrix are minimized. The Genomic SEM models are specified such that the SNP effect, defined by multiple traits, occurs at a level of a latent factor ($F_g$), and the model fit is assessed using model chi-square, Akaike information criterion and standardized root mean square. However, this method also provides evidence of heterogeneity between the phenotypes via the QSNP statistics, which show the extent to which the univariate regression effects of SNPs for each phenotype are explained by a common genetic factor. QSNP is a chi-square distributed statistic that can test whether SNPs act entirely through a common factor.

## Enrichment analyses

We used MAGMA v1.08 (as implemented in FUMA) to perform tissue and cell-type enrichment based on our multi-ancestry meta-analyses. First, we assessed whether our GGE GWAS was enriched for specific tissues from the GTEx database. Similarly, we assessed the enrichment of genes expressed in the brain at 11 general developmental stages, using data from the BrainSpan consortium. Next, we assessed whether GGE was associated with specific cell types, by cross-referencing two single-cell RNA-sequencing databases of human developmental and adult brain samples. The PsychENCODE database contains RNA-sequencing data from 4,249 human brain cells from developmental stages and 27,412 human adult brain cells[94]. The Zhong dataset (GSE104276) contains RNA-sequencing data from 2,309 human brain cells at different stages of development[95]. We performed FDR correction across datasets to assess which cell types were significantly associated with GGE. As a sensitivity analysis, we performed stratified LDSC with default settings using the cell-specific gene expression weights from the PsychENCODE consortium to compare GABAergic with glutamatergic neuron enrichment[96].

## Genetic overlap with other diseases

Using the FUMA web application, we searched the GWAS catalog for previously reported associations with $P < 5 \times 10^{-8}$ for SNPs at all 26 genome-wide significant loci.

Genetic correlations between 'all', FE and GGE and 18 other traits were computed with LDSC v1.01, using default settings. For these analyses, we used our European-only GWAS. Traits highlighted by the GWAS catalog analysis and/or those with established epilepsy comorbidity were prioritized and pursued provided recent summary statistics were available for public download (Supplementary Table 12). Although estimates are in general consistent between LDSC and LDAK[90], we decided to use LDSC as it is the more established method of the two for genetic correlations and used by almost all genetic correlation atlases and databases[97,98].

We used a recently described bivariate causal mixture model (MiXeR) v1.2.0 to quantify the polygenic overlap between GGE with the same 18 traits as assessed with LDSC. Bivariate MiXeR analyses estimate the total amount of causal SNPs underlying each trait, after which it assesses how many of these SNPs are shared between two traits. Notably, the number of overlapping SNPs is calculated regardless of the direction of effect. This makes it different from overall genetic correlation analyses such as LDSC, where overlapping SNPs with mixed directions of effect can cancel each other out, resulting in low genetic correlation. We used the same publicly available summary statistics as used for LDSC (Supplementary Table 12), after which bivariate MiXeR was run with default settings.

## Drug-repurposing analyses

We used a recently developed method that uses the GWAS for a disease to predict the relative efficacy of drugs for the disease[39]. We applied this method to 'all' epilepsy and GGE GWAS results, using (1) imputed gene expression data from the FUSION analyses, as described above, and (2) gene-based $P$ values from MAGMA (see above), with default settings. We predicted the relative efficacy of 1,343 drugs in total (Supplementary Data 8). We determined if our predictions correctly identify (area under the receiver operating characteristic curve) and prioritize (median rank) known clinically effective antiseizure drugs, as previously described[39]. We determined the statistical significance of drug identification and prioritization results by comparing the results to those from a null distribution generated by performing $10^6$ random permutations of the scores assigned to drugs.

## Reporting summary

Further information on research design is available in the Nature Portfolio Reporting Summary linked to this article.

## Data availability

The GWAS summary statistics data that support the findings of this study (for both multi-ancestry and European-only analyses) are publicly available at https://www.epigad.org/ and in the NHGRI-EBI GWAS Catalog at https://www.ebi.ac.uk/gwas/ (accession IDs: GCST90271608, GCST90271609, GCST90271610, GCST90271611, GCST90271612, GCST90271613, GCST90271614, GCST90271615, GCST90271616, GCST90271617, GCST90271618, GCST90271619 and GCST90271620). Individual-level GSA-MD v1.0 data for the Epi25 case samples and HKOS control samples are available in dbGaP/AnVIL under phs001489. v2.p2. GSA-MD v1.0 data for Genomic Psychiatry Cohort (GPC) control samples data will be made available in dbGAP/AnVIL under study phs002041. Individual-level SNP genotype data for other cohorts used as controls in the Epi25 analyses are accessible via an application through the THL Biobank portal (https://thl-biobank.elixir-finland. org/) for FINRISK, and in dbGaP/AnVIL under study accession numbers phs001642 (NIDDK IBDGC) and phs002018.v1.p1 (MGB Biobank) (see Supplementary Note for more details). Data relating to UK Biobank are available via the application to UK Biobank (https://www.ukbiobank. ac.uk/enable-your-research/apply-for-access). The FinnGen data can be accessed through the Fingenious services (https://site.fingenious.fi/ en/) managed by FINBB: release R6. The summary statistics of the Japanese GWAS in this study are publicly available from the National Bioscience Database Center (https://biosciencedbc.jp/en) under research ID: hum0014. We also accessed data from the following online database: www.DGidb.com (accessed on 26 November 2021). Source data are provided with this paper.

## Code availability

No custom code was used in this study. Publicly available software tools were used to perform genetic analyses and are referenced throughout the manuscript.

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

## Acknowledgements

Some of the data reported in this study were collected as part of a project undertaken by the ILAE and some of the authors are experts selected by the ILAE. Opinions expressed by the authors, however, do not necessarily represent the policy or position of the ILAE.

This study received support from Science Foundation Ireland (SFI; 16/RC/3948), cofunded under the European Regional Development Fund, the Research Unit FOR-2715 of the German Research Foundation (MN: NO755/6-1 and NO755/13-1), from Wellcome Trust (grant 084730), European Union's Seventh Framework Program (FP7/2007-2013) under grant agreement 279062 (EpiPGX), The Muir Maxwell Trust and the Epilepsy Society, UK and Fonds National de la Recherche Luxembourg (Research Unit FOR-2715, FNR grant INTER/DFG/21/16394868 MechEPI2) to P.M. and R. Krause. Part of this work was undertaken at University College London Hospitals, which received a proportion of funding from the NIHR Biomedical Research Centers funding scheme. Further support was received by a 'Vrienden WKZ' fund 1616091 (MING) to R. Stevelink and B.P.C.K., a National Health and Medical Research Council (NHMRC) of Australia Program Grant (1091593) to S.F.B. and I.E.S. and an NHMRC Investigator grant (APP1195236) to M.B. The Australian Government Research Training Program Scholarship (APP533086) provided by the Australian Commonwealth Government and the University of Melbourne supports K.L.O., a Wellcome Clinical Ph.D. Fellowship on the 4Ward North program (203914/Z/16/Z) supported D.L.-S., the UKRI MRC award MR/S02638X/1 and the NIHR Imperial Biomedical Research Center (BRC) support M.R.J., and Fundação de Amparo à Pesquisa do Estado de São Paulo (FAPESP), Brazil (grant 2013/07559-3) supported I.L.-C. The funding bodies had no role in the study design, data collection, analysis and interpretation, or in writing the manuscript.

We thank the Epi25 principal investigators, local staff from individual cohorts and all patients with epilepsy who participated in research studies at local centers for making possible this global collaboration and resource to advance epilepsy genetics research. This work is part of the Centers for Common Disease Genomics (CCDG) program, funded by the National Human Genome Research Institute (NHGRI), The Eunice Kennedy Shriver National Institute of Child Health and Human Development and the National Heart, Lung and Blood Institute (NHLBI). CCDG-funded Epi25 research activities at the Broad Institute, including genomic data generation in the Broad Genomics Platform, were supported by NHGRI grant UM1 HG008895 (PIs: E. Lander, S. Gabriel, M. Daly, S. Kathiresan). The Genome Sequencing Program efforts were also supported by NHGRI under grant 5U01HG009088-02. The content is solely the responsibility of the authors and does not necessarily represent the official views of the National Institutes of Health. We thank the Stanley Center for Psychiatric Research at the Broad Institute for supporting the genomic data generation efforts as well as the aggregation of control samples and cohorts to contribute to the Epi25 GWAS analyses. In particular, the Genomic Psychiatry Cohort controls were genotyped on the GSA-MD v1.0 by the Broad Genomics Platform with funding from NIH grant U01MH105641 and the Stanley Center for Psychiatric Research, Broad Institute of MIT and Harvard. The FINRISK controls were part of the FINRISK studies supported by the THL (formerly KTL: National Public Health Institute) through budgetary funds from the government, with additional funding from institutions such as the Academy of Finland, the European Union, ministries and national and international foundations and societies to support specific research purposes. The collection of the Hong Kong Osteoporosis Study (HKOS) control samples was funded by the Bone Health Fund and Research Grants Council—Early Career Scheme (project 27100416). Other control datasets included IBD NIDDK and samples from the Mass General Brigham (MGB) Biobank available from dbGaP under study accession number phs002018.v1.p1.

We acknowledge the participants and investigators of the FinnGen study. The FinnGen project is funded by two grants from Business Finland (HUS 4685/31/2016 and UH 4386/31/2016) and the following industry partners: AbbVie, AstraZeneca UK, Biogen MA, Bristol Myers Squibb (and Celgene Corporation & Celgene International II Sàrl), Genentech, Merck Sharp & Dohme Corp,

Pfizer, GlaxoSmithKline Intellectual Property Development, Sanofi US Services, Maze Therapeutics, Janssen Biotech, Novartis AG and Boehringer Ingelheim. The following biobanks are acknowledged for delivering biobank samples to FinnGen: Auria Biobank (www.auria.fi/biopankki), THL Biobank (www.thl.fi/biobank), Helsinki Biobank (www.helsinginbiopankki.fi), Biobank Borealis of Northern Finland (https://www.ppshp.fi/Tutkimus-ja-opetus/Biopankki/Pages/Biobank-Borealis-briefly-in-English.aspx), Finnish Clinical Biobank Tampere (www.tays.fi/en-US/Research_and_development/Finnish_Clinical_Biobank_Tampere), Biobank of Eastern Finland (www.ita-suomenbiopankki.fi/en), Central Finland Biobank (www.ksshp.fi/fi-FI/Potilaalle/Biopankki), Finnish Red Cross Blood Service Biobank (www.veripalvelu.fi/verenluovutus/biopankkitoiminta) and Terveystalo Biobank (www.terveystalo.com/fi/Yritystietoa/Terveystalo-Biopankki/Biopankki/). All Finnish biobanks are members of BBMRI.fi infrastructure (www.bbmri.fi). Finnish Biobank Cooperative—FINBB (https://finbb.fi/)—is the coordinator of BBMRI-ERIC operations in Finland. The Finnish biobank data can be accessed through the Fingenious services (https://site.fingenious.fi/en/) managed by FINBB.

## Author contributions

Data analysis: Analytical design, imputation. O.M.A., M.B., C. Campbell (lead analyst), G.L.C., S.C. (lead analyst), Y.-C.A.F., E. Hassanin, B.P.C.K., R. Krause (data management), D. Lal, C.L., N.M., M.N., K.L.O., R. Stevelink (lead analyst). Data generation and quality control and management: L.B., D.R.B., J.P.B., R.J.B., G.L.C., F. Cerrato, S.S.C., C. Churchhouse, C. Cusick, Y.-C.A.F., N.G., H. Hakonarson, E.L.H., I.H., D.P.H., D.K., B.P.C.K., R. Krause., D. Lal, Z.L., C.L., I.L.-C., P.M., N.M., B.M.N., P.N., S.P., T. Sander, D.S., R. Stevelink, F. Zara, W.Z. Analysis coordination: G.L.C. (Cochair), B.P.C.K. (Cochair). External data resources and analysis: UK Biobank: C. Campbell, D.L.-S., R.H.T. BioBank Japan: Y. Kamatani, M. Kanai, M. Kato, Y.O. FinnGen: M.J.D., H.O.H., R. Kälviäinen, M.I.K., A. Palotie. deCODE genetics: S.M., E.Ó., H. Stefansson, K.S., U.U. Writing committee: O.M.A., M.B., S.F.B., C. Campbell, G.L.C., S.C., B.P.C.K., K.L.O., R. Stevelink (wrote first draft). Strategy committee: L.B., S.F.B. (Chair), R.J.B., G.L.C., H. Hakonarson, E.L.H., M.R.J., R. Kälviäinen, B.P.C.K., R. Krause, P. Kwan, D. Lal, H.L., Q.S.L., I.L.-C., D.H.L., T.J.O'B., S.M.S. Phenotyping committee: C.D., D.J.D., W.S.K., P. Kwan, D.H.L. (Chair), A.G.M., P. Striano. Governance committee: S.F.B., A. Compston, A.-E.L., D.H.L. Patient recruitment and phenotyping: B.A.-K., Z.A., E.A., A. Anderson, J.A., D.M.A., G.A., P.A., A. Avbersek, M.D.B., G.B., S.B., C. Barba, K. Barboza, F. Bartolomei, T. Bast, T. Baumgartner, B. Baykan, N. Bebek, A.J.B., F. Becker, C.A.B., B. Berghuis, S.F.B., A.B., C. Bianchini, F. Bisulli, I. Blatt, I. Borggraefe, C. Bosselmann, V. Braatz, K. Brockmann, R.J.B., R.M.B., H.C., E.C., L.C., C. Canavati, G.D.C., B.C., C.B.C., F. Chassoux, K.C., I.-J.C., S.-K.C., P.O.C., A.J.C., A. Coppola, M.C., P.C., J.J.C., L.K.D., G.-J.d.H., N.D., C.D., P.D., O.D., L.D.V., D.J.D., V.D., C.P.D., H.E.-N., C.E.E., C.A.E., A.Faucon, L.Ferguson, T.N.Ferraro, L.Ferri, M.Feucht, M.Fitzgerald, B.Fonferko-Shadrach, F.Fortunato, S.Franceschetti, J.A.F., E.F., M.G., A. Gambardella, E.B.G., T. Giangregorio, L.G., T.

Glauser, E.G., A. Goldman, T. Granata, D.A.G., R.G., K.F.H., K.H., M.H., I.H., C.H., S.H., E. Hirsh, H. Hjalgrim, D.H., P.-C.H., M.I., L.L.I., Y.I., A.I., J.J.-K., L.J., M.R.J., R. Kälviäinen, M. Kanaan, A.-M.K., B.K., S.M.K., D.K.-N.T., J.K., Y. Kesim, N.K.-Z., C.K., H.E.K., K.M.K., G. Kluger, S.K., R.C.K., A.D.K., A.K., I. Kousiappa, M. Krenn, H.K., I. Krey, W.S.K., G. Kurlemann, G. Kuzniecky, P. Kwan, A. Labate, A. Lacey, S. Lauxmann, S.L.L., A.-E.L., J.R.L., H.L., G.L., N.L., Q.S.L., L. Licchetta, K.-L.L., D. Lindhout, T.L., I.L.-C., D.H.L., C.H.T.L., F.M., A.G.M., C.M.M., D.M., R.M., R.S.M., M.M., B.M., L.M., H.M., K.M.-S., I.M.N., W.N., B.N., C.R.J.C.N., T.J.O'B., Ç.Ö., S.S.P., E.P., M. Pendziwiat, W.O.P., R.P., T.P., A. Poduri, F. Pondrelli, R.H.W.P., M. Privitera, A. Rademacher, R.R., F. Ragona, S. Rau, M.I.R., B.M.R., P.S.R., S. Rhelms, A. Riva, F. Rosenow, P.R., A. Saarela, L.G.S., J.W.S., T. Sander, M.S., T. Scattergood, S.C.S., C.J.S., I.E.S., B.S., S.S., S.S.-B., A.S.-B., P. Scudieri, B.R.S., J.J.S., G.J.S., S.M.S., M.C.S., P.E.S., A.C.M.S., M.R.S., B.J.S., U.S., W.C.S., C.S., P. Striano, H. Stroink, A. Strzelczyk, R. Surges, T. Suzuki, K.M.T., R.S.T., G.A.T., E.T., L.L.T., O.T., P. Tinuper, M.T., P. Topaloglu, R.T., M.-H.T., B.T., D.T., A.U., P.V., L.V., A.v.b., A.V., E.P.G.V., F.V., S.v.B., R.v.W., R.G.W., Y.G.W., S. Weckhuysen, J.W., M. Weller, P.W.-W., M. Wolff, S. Wolking, D.W., K.Y., Z.Y., E.Y., S.Z., F. Zahnert, F. Zimprich, G.Z., Q.Z.A. Control cohorts: L.C.B., C.-L.C., J.G.E., A. Franke, H. Hakonarson, Y.-L.L., J.L.M., A.M.M., M.M.N., A. Palotie, F. Pangilinan, C.N.P., M.T.P., P. Sham, H. Stroink, G.N.T., W. Yang. Consortium coordination: K.L.O.

## Competing interests

G.L.C. is in receipt of research funding from Congenica and Janssen Pharmaceuticals and has conducted consultancy for Ono Pharmaceuticals. S.F.B. received funding from UCB Pharma and Eisai and has been a consultant for Praxis Precision Medicines and Sequiris. Q.S.L. is an employee of Janssen Research & Development, LLC and a shareholder in Johnson & Johnson, which is the parent company of the Janssen companies. B.M.N. currently serves as a member of the scientific advisory board at Deep Genomics and Neumora (previously RBNC) and as a consultant for Camp4 Therapeutics. S.P. is an employee and shareholder of AstraZeneca. U.U., S.M., H. Stefansson and K.S. are employees of deCODE genetics/Amgen.

## Additional information

**Correspondence and requests for materials** should be addressed to Samuel F. Berkovic, Gianpiero L. Cavalleri or Bobby P. C. Koeleman.

# Reporting Summary

## Statistics

For all statistical analyses, confirm that the following items are present in the figure legend, table legend, main text, or Methods section.

| n/a | Confirmed | |
|---|---|---|
| ☐ | ☒ | The exact sample size (*n*) for each experimental group/condition, given as a discrete number and unit of measurement |
| ☒ | ☐ | A statement on whether measurements were taken from distinct samples or whether the same sample was measured repeatedly |
| ☐ | ☒ | The statistical test(s) used AND whether they are one- or two-sided *Only common tests should be described solely by name; describe more complex techniques in the Methods section.* |
| ☐ | ☒ | A description of all covariates tested |
| ☐ | ☒ | A description of any assumptions or corrections, such as tests of normality and adjustment for multiple comparisons |
| ☐ | ☒ | A full description of the statistical parameters including central tendency (e.g. means) or other basic estimates (e.g. regression coefficient) AND variation (e.g. standard deviation) or associated estimates of uncertainty (e.g. confidence intervals) |
| ☐ | ☒ | For null hypothesis testing, the test statistic (e.g. *F*, *t*, *r*) with confidence intervals, effect sizes, degrees of freedom and *P* value noted *Give P values as exact values whenever suitable.* |
| ☒ | ☐ | For Bayesian analysis, information on the choice of priors and Markov chain Monte Carlo settings |
| ☒ | ☐ | For hierarchical and complex designs, identification of the appropriate level for tests and full reporting of outcomes |
| ☐ | ☒ | Estimates of effect sizes (e.g. Cohen's *d*, Pearson's *r*), indicating how they were calculated |
| | | *Our web collection on statistics for biologists contains articles on many of the points above.* |

## Software and code

Policy information about availability of computer code

| | |
|---|---|
| Data collection | Genotyping was performed at various sites as detailed in the Methods and Supplementary Table 1. Genotypes were harmonized to the Haplotype Reference Consortium (HRC) panel v1.1 using tools available from https://www.well.ox.ac.uk/~wrayner/tools/ prior to imputation using one of the two following imputation servers: 1. Wellcome Sanger Institutes imputation server: https://imputation.sanger.ac.uk 2. Michigan Imputation server: https://imputationserver.sph.umich.edu/ |
| Data analysis | No custom code was used in this study. We used the following public and freely available software tools to preform the reported analyses: Eagle v2.4.1 for haplotype phasing; PBWT v3.1 for imputation; Plink v1.9 for genotype data manipulation/QC; Saige v0.38 for generalized mixed models; METAL v2020-05-05 for meta-analyses; Metasoft v2.0.0 for generating m-values for PM-Plots; R v3.6.0 for generating Manhattan and correlation plots; ASSET v2.2.0 for pleiotropy detection; CookHLA v1.0.1 for HLA imputation; HLA Analysis Toolkit (HATK) v1.2 for HLA association analysis ANNOVAR v2017-07-17 for SNP annotation; MTAG v1.0.8 for multi-trait analysis; FUMA v1.3.8 for gene mapping; MAGMA v1.08 for gene set analyses; |

FUSION v3 for TWAS;
GWAMA v2.2.2 for sex-specific GWAS
LDAK v5.2 for SNP-based heritability analyses;
MiXeR v1.2.0 for causal mixture models;
LDSC v.1.01 for cross-train genetic correlations;
Relevant references are listed throughout the manuscript for all above stated tools and software.

For manuscripts utilizing custom algorithms or software that are central to the research but not yet described in published literature, software must be made available to editors and reviewers. We strongly encourage code deposition in a community repository (e.g. GitHub). See the Nature Portfolio guidelines for submitting code & software for further information.

## Data

Policy information about availability of data

All manuscripts must include a data availability statement. This statement should provide the following information, where applicable:

- Accession codes, unique identifiers, or web links for publicly available datasets
- A description of any restrictions on data availability
- For clinical datasets or third party data, please ensure that the statement adheres to our policy

The GWAS summary statistics data that support the findings of this study (for both multi-ancestry and European-only analyses) are publicly available at https://www.epigad.org/ and in the NHGRI-EBI GWAS Catalog at https://www.ebi.ac.uk/gwas/ (accession IDs: GCST90271608, GCST90271609, GCST90271610, GCST90271611 ,GCST90271612, GCST90271613, GCST90271614, GCST90271615, GCST90271616, GCST90271617, GCST90271618, GCST90271619 & GCST90271620). Individual-level GSA-MD v1.0 data for the Epi25 case samples and HKOS control samples are available in dbGaP/AnVIL under phs001489.v2.p2. GSA-MD v1.0 data for Genomic Psychiatry Cohort (GPC) control samples data will be made available in dbGAP/AnVIL under study phs002041. Individual-level SNP genotype data for other cohorts used as controls in the Epi25 analyses are accessible via an application through the THL Biobank portal (https://thl-biobank.elixir-finland.org/) for FINRISK, and in dbGaP/AnVIL under study accession numbers phs001642 (NIDDK IBDGC) and phs002018.v1.p1 (MGB Biobank) (see Supplementary information for more details). Data relating to UKBiobank is available via application to UKBiobank (https://www.ukbiobank.ac.uk/enable-your-research/apply-for-access). The FinnGen data can be accessed through the Fingenious services (https://site.fingenious.fi/en/) managed by FINBB: release R6. The summary statistics of the Japanese GWAS in this study are publicly available from the National Bioscience Database Center (https://biosciencedbc.jp/en) under research ID: hum0014. We also accessed data from the following online database: www.DGidb.com (accessed on 26-11-2021).

## Human research participants

Policy information about studies involving human research participants and Sex and Gender in Research.

| Reporting on sex and gender | Self-identified gender was collected upon recruitment of both cases and controls. Biological sex was subsequently determined genetically and used for downstream analyses. |
|---|---|
| Population characteristics | Diagnoses: 29944 patients with epilepsy, 52538 controls<br>Genotypic ancestry (cases & controls): 69995 European, 5306 Asian, 7181 African<br>Sample age data was not available. |
| Recruitment | Case and control samples were recruited from tertiary hospital and academic research centres. All cases were diagnosed with epilepsy syndrome according to the same international guidelines and classification system, however, it is possible that the application of diagnostic criteria across cohorts may slightly differ. This ascertainment bias may have resulted in a reduction to the overall power of the study and the generalizability of results. |
| Ethics oversight | All contributing case and control sites collected samples following local IRB/ethics committee approval. A full list of approval bodies can be found in Supplementary Table 1. |

Note that full information on the approval of the study protocol must also be provided in the manuscript.

## Field-specific reporting

Please select the one below that is the best fit for your research. If you are not sure, read the appropriate sections before making your selection.

☒ Life sciences    ☐ Behavioural & social sciences    ☐ Ecological, evolutionary & environmental sciences

For a reference copy of the document with all sections, see nature.com/documents/nr-reporting-summary-flat.pdf

## Life sciences study design

All studies must disclose on these points even when the disclosure is negative.

| Sample size | Sample size was not predetermined, however, we note that this study is almost twice the size of the previous largest epilepsy GWAS published in 2018. |
|---|---|
| Data exclusions | We excluded poorly genotyped SNPs and outlier samples according to the various QC parameters which are described in our methods. |

| | |
|---|---|
| Replication | As this is the largest study of common variants in epilepsy to date we did not have additional samples to replicate the significant variants in this study. However we have demonstrated reproducibility of our previous study findings and expect to reproduce this study findings with the addition of new samples in future work. |
| Randomization | There was no randomization in this study. Cases were grouped according to electroclinical phenotypes in epilepsy while controls were unscreened population samples. |
| Blinding | Samples were coded at collection and phenotypes were collected by separate individuals from analysts, preventing analysts from making genotype-phenotype identification of a study participant. |

# Reporting for specific materials, systems and methods

We require information from authors about some types of materials, experimental systems and methods used in many studies. Here, indicate whether each material, system or method listed is relevant to your study. If you are not sure if a list item applies to your research, read the appropriate section before selecting a response.

## Materials & experimental systems

| n/a | Involved in the study |
|---|---|
| ☒ | Antibodies |
| ☒ | Eukaryotic cell lines |
| ☒ | Palaeontology and archaeology |
| ☒ | Animals and other organisms |
| ☒ | Clinical data |
| ☒ | Dual use research of concern |

## Methods

| n/a | Involved in the study |
|---|---|
| ☒ | ChIP-seq |
| ☒ | Flow cytometry |
| ☒ | MRI-based neuroimaging |

