## [Peer Review File · Nature Genetics]

Peer Review Information

Manuscript Title: Professor Samuel (F) Berkovic Dr Gianpiero (L) Cavalleri Dr Bobby (P C) Koeleman

Corresponding author name(s): Genome-wide association meta-analysis of over 29,000 people with epilepsy identifies 26 risk loci and subtype-specific genetic architecture

Reviewer Comments & Decisions:

Decision Letter, initial version:
--

10th August 2022

Dear Dr. Berkovic,

Your Article entitled "Genome-wide meta-analysis of over 29,000 people with epilepsy reveals 26 loci and subtype-specific genetic architecture" has been seen by two referees. You will see from their comments below that, while they find your work of interest, they have raised several relevant points. We are interested in the possibility of publishing your study in Nature Genetics, but we would like to consider your response to these points in the form of a revised manuscript before we make a final decision on publication.

To guide the scope of the revisions, the editors discuss the referee reports in detail within the team with a view to identifying key priorities that should be addressed in revision, and sometimes overruling referee requests that are deemed beyond the scope of the current study. In this case, we particularly ask that you address all technical queries related to the primary association analyses and their interpretation, extend the subtype-specific and genetic correlation analyses where feasible, and revise the presentation throughout to include additional details where needed. We hope you will find this prioritized set of referee points to be useful when revising your study. Please do not hesitate to get in touch if you would like to discuss these issues further.

We therefore invite you to revise your manuscript taking into account all reviewer and editor comments. Please highlight all changes in the manuscript text file. At this stage, we will need you to upload a copy of the manuscript in MS Word .docx or similar editable format.

*2) If you have not done so already please begin to revise your manuscript so that it conforms to our Article format instructions, available [here](http://www.nature.com/ng/authors/article_types/index.html). Refer also to any guidelines provided in this letter.

[redacted]

We hope to receive your revised manuscript within 8-12 weeks. If you cannot send it within this time, please let us know.

Sincerely,
Kyle

Kyle Vogan, PhD
Senior Editor
Nature Genetics
<https://orcid.org/0000-0001-9565-9665>

Referee expertise:

Referee #1: Genetics, neurology

Referee #2: Genetics, neurodevelopmental disorders, psychiatric diseases

Reviewers' Comments:

Reviewer #1:
Remarks to the Author:

Thank you for giving me the opportunity to review the manuscript "Genome-wide meta-analysis of over 29,000 people with epilepsy reveals 26 loci and subtype-specific genetic architecture" by the International League Against Epilepsy Consortium on Complex Epilepsies. This work has been a huge collaborative effort and the sample size has increased considerably compared to the previous analysis from the consortium. The subtype analyses are really informative. This work reports several novel loci for the epilepsies, in particular for GGE. Extensive follow-up analyses were performed to prioritise genes underlying the GWAS signal. There are however some issues to address.

Major:

1) The authors elegantly phrase their lack of replication "this work highlights the challenges..." in the discussion. While I appreciate their point that the main analysis cohorts are more accurately characterised, shouldn't we expect at least replication for the "all epilepsy" phenotype, given that the biobank data has approximately the same effective sample size? What are the p-values for the biobank data in the top-loci from the main analysis? Is there any way the authors can provide (suggestive) evidence that the lack of replication is indeed due to a milder or more heterogeneous phenotype?

2) There seems to be a lack of information on heterogeneity: to fully appreciate the results, I feel QQ-plots for the full GWAS, and heterogeneity measures/forest plots for the top-hits should be provided. Additionally, I would be interested to see how the effects the biobank data (locally) compare to the main analysis data (see also point 1 above).

3) At many points in the manuscript, information is lacking or only available by reading other work. Below there are some examples, but there are likely others. Specifically,

a) the authors refer to their previous publication regarding included cohorts, classification of epilepsy - both crucial pieces of information - and in sentences like " .. with the same population prevalences as in our previous study". This information is in supplementary table 7, why not refer to that? It would be much easier for the reader to appreciate this manuscript if the information was included.

b) Explanation of methods is extremely brief. There is no need to repeat the papers that introduce the methodology, but the rationale for some of the analyses could be explained better. As examples: why do the authors use both LDSC and LDK for heritability estimates and why are their differences not discussed? Why running both TWAS and SMR if they test the same hypotheses? "Summary-data-based Mendelian Randomization (SMR) v1.03 is an additional method to assess the association between epilepsy and expression of specific genes." Another example: "...after which bivariate MiXeR was run with default settings" later in the manuscript, MiXeR is introduced as "The total amount of causally associated variants (i.e., variants with nonzero additive genetic effect) underlying epilepsy risk was calculated by a causal mixture model (MiXeR)." What is tested in such a bivariate model? "This sex difference is further corroborated by significant sex-heterogeneity ($p=1.54 \times 10^{-8}$) and gender-differentiation ($p=5.6 \times 10^{-9}$)." What exactly is tested in each of these cases? The methods section does not help.

c) Supplementary Tables/Data are not readable on their own. Why not put all the data tables in one document and add an index, plus provide explanations for the column headers at the top of each table?

4) Drug repurposing: The conclusions on drug repurposing seem a bit circular. If done using the same procedure used by Mirza et al. (2021), then there are two parameters (gene-list length of AC score and the weight used in the AC and FM average) that are optimized for predicting known anti-epilepsy medications (ASMs). It is understandable that the currently approved medications for epilepsy are the best treatments we have, and therefore, can be used to optimize a model. However, I don't think one can then conclude the drug repurposing results are validated based on finding ASMs using this method, since predicting ASMs has been built into the model. In addition, is a permutation test using randomly assigned scores a fair comparison, given that the ability to predict ASM scores has been built into the model? To fully appreciate the drug repurposing results, median ranks and AUROC scores of all drug groups (including ASMs) should be provided.

5) Results "Genomic inflation was comparable to our previous GWAS and all linkage-disequilibrium score regression (LDSR) intercepts were lower than in our previous GWAS (Supplementary table 6) suggesting that the signals are primarily driven by polygenicity, rather than by confounding or population stratification." At first glance, the intercept of 1.10 does not seem to be extremely high. But then (only when reading the methods) I discovered that these analyses were done in the European sample only. As a general point, it should be made clear through the manuscript (and in table/figure captions) whether the shown results stem from the trans-ancestry analyses or from the European-only analysis. In light of this, a LDSC intercept of 1.10 does not seem to exclude confounding such as population stratification. Following a commonly used approach and computing the ratio $(\text{LDSC intercept} - 1) / (\text{mean Chi}^2 - 1)$ using supplemental table 6 a substantial proportion (37%, 48% and 11%) of the inflation might be due to some form of bias (e.g. stratification, cryptic relatedness or other confounding factors). Please comment.

Other points:

6) Abstract: "SNP-based heritability analyses show that common variants substantially close the missing heritability gap for GGE". I assume this refers to the liability scale heritability but it is hard to judge without knowing the twin-heritability. In addition, in the introduction "Nevertheless, previous genetic studies of common epilepsies have explained only a few percent of this common genetic

variant, or SNP-based, heritability". What is this percentage? Please add these numbers to the manuscript.

7) Introduction, page 2: "Here, we report the third epilepsy GWAS meta-analysis..." the authors probably mean the ILAE third GWAS, there are more out there (e.g. Song et al., 2021). In addition, the caption of Figure 2 states: "Loci were classified as novel or replication according to the genome-wide significant results of previous GWAS publications" What are these publications? Related: in the results: "The selected traits had either, or a combination of 1) epilepsy as a common comorbidity or 2) pleiotropic loci shared with epilepsy". Pleiotropic loci based on what? From the methods I guess that it is the GWAS catalog?

8) Results, page 2: "We further conducted meta-analyses in subjects of European ancestry of the well-defined GGE subtypes ... lesion-negative focal epilepsy". I would appreciate a mention of the sample sizes here. Were controls shared in these analyses?

9) Results, page 2: Why was functional annotation only done for the GGE analyses?

10) Results, page 3: "Our 'all epilepsy' meta-analysis revealed four genome-wide significant loci, of which two were novel (Figure 1)." The authors then move forward and discuss a sub-threshold locus. Why not discuss the location/function of the two novel loci?

11) Results, page 5: "Interrogation of the Drug Gene Interaction Database (DGIdb) showed that 13 of the 29 genes are targeted by a total of 214 currently licensed drugs (Supplementary data 6)." It would be interesting to discuss what these drugs are licensed for, and/or refer to Supplementary Table 8 (this is where I found the information, but that was not mentioned in the text until much later. Alternatively, this paragraph could be removed.

12) Results, page 6; Fig 2: Could the authors provide a measure of the reliability of the highest scoring genes? i.e. is this by far the highest scoring gene, or could there be other genes in this locus that score e.g. one less? In addition, are these 10 criteria truly independent - in the sense that they capture a different aspect of functionality? I can imagine with all the underlying data sources in the methods used, there might be overlap. If this is not the case, it would be good to state this explicitly in the methods section.

13) Supplementary Data 5: why are there NAs in the table?

14) Results, page 8: The rationale for the analysis between ASD, intelligence and epilepsies is not clear (or rather, receives a lot of attention compared to other parts). If I understand the reasoning correctly, the authors attempt to show that the percentage of discordant/concordant directions of effect are similar associating epilepsy with either ASD or intelligence. This could very well be, but it does not explain why intelligence then shows a global association and ASD does not. MiXeR provides a different type of results compared to LDSC, so I do not see why such a comparison would imply overlap between ASD and GGE. Perhaps the authors wish to run the MiXeR analyses for all phenotypes and compare results; then the difference in interpretation between LDSC and MiXeR analyses should be discussed. In addition, couldn't the lack of correlation be a consequence of the limited sample size for ASD?

15) Discussion, page 10: "It would seem that focal epilepsies, as a group, are far more heterogeneous

than GGE". Could an alternative explanation be that this phenotype is more polygenic?

16) Methods, page 12: Why the difference in QC criteria for the new cohorts and Epi25? Specifically, the difference in INFO score cut-off is striking. As a side note, methods are describing QC procedures with either filtering criteria or "QCed for" and these criteria have opposite signs, which can be confusing. Please harmonise.

17) Methods, page 15: "Brain expression: we calculated mean expression of all brain and non-brain tissues based on data from the Genotype-Tissue Expression (GTEx) project v8 and assessed if the average brain tissue expression was higher than the average expression in non-brain tissues." This criterion is not clear to me as it is not mentioned how the GWAS results come into this. Is it tissue-specificity - similar to tissue-specificity implemented in FUMA?

Reviewer #2:

Remarks to the Author:

The authors present updated results from GWASs in epilepsy subtypes with secondary analyses further characterizing their findings and their potential implications regarding clinical ramifications (e.g. by repurposing existing drugs). The presented analyses and results are certainly interesting and well described (albeit some sections would benefit from slightly more details). However, the biggest criticism (see below) can probably be summarized as "missed opportunities" as current GWAS papers in complex traits genetics often include more analytical approaches (especially when it comes to heterogeneous phenotypes such as the ones presented here). Without these additional analyses, this is still a remarkable effort and manuscript but likely less interesting to a broader readership. It obviously can't be ruled out that such analyses are earmarked for additional manuscripts, but this reviewer would support inclusion in the current manuscript.

Major points:

(1) Results from the presented analyses (and previous GWAS) indicate heterogeneity in the different epilepsy subtypes when it comes to the underlying genetic architecture. The authors take only modest interest in analyzing the heterogeneity given the results. Granted, the low sample size might fuel this in the different sub-types but at least for some of their sub-types sample size seems high enough to warrant such an analysis. For example, a case-case GWAS between FE and GGE could help shed light on the difference in genetic architecture between the two. Applying tools such as GW-SEM (if individual-level data is available across datasets) or at least genomic SEM would potentially provide further insights (e.g., concerning a common genetic factor across subtypes). Given the heterogeneity, it might have also made sense to use such tools to perform an overall analysis (with presumably added power compared to the crude meta-analysis for the "all epilepsy" analysis). Tools such as MTAG could have helped (with obvious limitation on small sample size, see above) to increase the "virtual" sample size for their subtypes. Tools such as METASOFT and their package to produce P-M plots could have helped to characterize the heterogeneity (<https://journals.plos.org/plosgenetics/article?id=10.1371/journal.pgen.1002555>). Again, it is appreciated that sample size might be an issue here for some of the subtypes, however, for others, it is not and some of the mentioned tools (e.g. METASOFT) could have helped to answer questions around sample size limitations.

(2) Along the lines of the comment above the relationship between the currently presented samples (and results) in the main analyses and biobanks should probably be further characterized. It is by now a well-known observation that different ascertainment schemes (especially in the context of incidence vs. prevalence samples etc.) lead to different “cuts” of the phenotypic spectrum of a disorder being on display and, as such, differences in genetic architecture being identified (with often misaligned heritability figures, etc.). At a bare minimum, the currently presented observation would point to potential ramifications for the interpretation of their results in the main analyses (and generalizability).

(3) In the interpretation and analyses of genetic correlations between autism spectrum disorders (ASD), intelligence, and epilepsy sub-types, the authors emphasize the observation that despite low r_G between ASD and epilepsy subtypes, there could be potential overlap between the genetic architecture for both conditions. This is important to note but falls short of further issues: among others, the shared genetic architecture between epilepsy subtypes and ASD might only reflect overlap with parts of the phenotypic spectrum of the condition (i.e., the part that concerns intellectual disability). Furthermore, other aspects of the genetic correlation with the traits in Figure 3 went unstudied (in this comparison, for example, the relationship between ADHD, ASD, intelligence/cognition/EA, and epilepsy subtypes (with links between ADHD and ASD and stronger r_G between epilepsy subtypes and ADHD compared to ASD). A more detailed characterization of the genetic relationship with febrile seizures and, in general, with phenotypes for which there seems to be a discordance (either because of sample size or real genetic differences) in r_G for epilepsy subtypes would likely have helped to further characterize the genetic architecture for epilepsy. This could, for example, also have included LAVA-like analyses (i.e., analyses of regional r_G).

(4) Sex-stratified analyses have the inherent feature of reducing sample size significantly. While that might not lead to differences in reduction of sample size for the two sexes (i.e., their GWAS sample size are the same), it still opens the possibility of random fluctuation in GWAS results. This is even likely at lower sample sizes that only capture part of the genetic architecture of a given trait. To truly identify loci that have different influences on the genetic architecture, one would have to use different tools or approaches. Case-case analyses (see above) might have helped highlight genetic architecture differences. Additionally, random subsampling of the sample (i.e., drawing random samples at the same sample size as stratified GWASs) might have added insights and helped answer the question of whether some loci are associated with one sex but not the other.

(5) When comparing current results to their previous GWAS, the authors generally only conclude that they identified additional loci. However, they fail to assess whether they also “lost” loci. It is to be supported that they make efforts to refine their previous findings (e.g., when it comes to the likely causal genes). Still, it seems equally important to highlight which associations are no longer supported. I might have overlooked this, but what happened, for example, to the association on chromosome 4 close to GABRA2 for GGE or focal epilepsy finding on chromosome 2? The latter is still present in the “all epilepsy” analysis. Still, these differences might point to changes in the phenotypic spectrum present in the samples, and analyses comparing results from previous samples and the new ones added here could help interpret the results. That might have been published elsewhere, but a small recap might be helpful.

Minor points:

(1) The authors claim (page 8, sex-specific analysis) that two lead SNPs from their analyses are not in LD (rs72845653, rs11191156). They base this observation on a r^2 value of 0.05. Unfortunately, they fail to appreciate that at the same time D' is at 0.87, and hence the markers are, in fact, in LD, albeit not in perfect LD (which is also not expected given the difference in MAF). See also here: https://ldlink.nci.nih.gov/?var1=rs72845653&var2=rs11191156&pop=CEU%2BTSI%2BFIN%2BGBR%2BIBS&genome_build=grch37&tab=ldpair).

(2) It seems that the authors used analytical tools (e.g., LDSC, MAGMA, etc.) that require reference samples. However, in some cases, there seems to be no information available about which reference samples were used. For the instances where it was denoted (for the LDK-based heritability analyses), it might have been interesting to see what different reference samples, including representation from regions that have been present in the study sample, would have yielded (given the heterogeneity of the study sample with samples from Finland, Norway, Austria, Switzerland, etc. with sometimes significant differences in contribution to the different subtype GWASs).

Manuel Mattheisen

Author Rebuttal to Initial comments

Point by point response to referees

Reviewers' Comments:

Reviewer #1:

Remarks to the Author:

Thank you for giving me the opportunity to review the manuscript "Genome-wide meta-analysis of over 29,000 people with epilepsy reveals 26 loci and subtype-specific genetic architecture" by the International League Against Epilepsy Consortium on Complex Epilepsies. This work has been a huge collaborative effort and the sample size has increased considerably compared to the previous analysis from the consortium. The subtype analyses are really informative. This work reports several novel loci for the epilepsies, in particular for GGE. Extensive follow-up analyses were performed to prioritise genes underlying the GWAS signal. There are however some issues to address.

Major:

1) The authors elegantly phrase their lack of replication "this work highlights the challenges..." in the discussion. While I appreciate their point that the main analysis cohorts are more accurately

characterised, shouldn't we expect at least replication for the "all epilepsy" phenotype, given that the biobank data has approximately the same effective sample size? What are the p-values for the biobank data in the top-loci from the main analysis? Is there any way the authors can provide (suggestive) evidence that the lack of replication is indeed due to a milder or more heterogeneous phenotype?

Author response: We thank the reviewer for highlighting the need for further exploration of the consistency of signal between the main and biobank analysis.

We now provide forest plots, comparing 1) the biobanks, 2) ILAE_2, 3) EPI25 and 4) the current main GWAS meta analyses, ILAE_3 (see results section and **Supplementary figure 23**).

These forest plots clearly illustrate in the biobanks, consistency in direction of effect, albeit with smaller effect sizes, for all top SNPs in the main all epilepsy (**Supplementary Figure 23**) and all but one SNP in the GGE meta-analyses (**Supplementary Figures 2 and 3**).

The p-values for the four top level loci in the all-epilepsy analysis are: 3.8×10^{-5} for rs59237858 (*SCN1A*), 0.001 for rs13032423 (*BCL11A*), 0.65 for rs4744696 (*RORB*) and 0.0150 for rs3740422 (*SCEL*).

Similarly, we show strong, but imperfect, genetic correlations between the biobanks and our main GWAS (**Supplementary table 14**).

Considered together, we feel these additional analysis/results suggest that the same genetic risk/protection variants underlie the biobank and main epilepsy GWAS, but the signal in the biobanks is weaker. We hypothesise in the discussion how the weaker biobank signal could be due to ascertainment in biobanks for milder epilepsy, combined with less well characterised and more heterogeneous phenotypes in the biobanks.

These new forest plots are described in the result section (page 3) which now reads: "*Forest plots and P-M plots of these signals show they appear consistent across all four GGE subphenotypes, with some exceptions (Supplementary figures 2 - 4).*" and section on page 9 entitled "GWAS in epilepsies ascertained through biobanks..." now reads: "*Forest plots showed a consistent direction of effect between the biobanks and our primary GWAS for all biobank-genotyped genome-wide significant top SNPs of the 'all epilepsy' GWAS and for all but one GGE top SNP (Supplementary figures 3 and 23).*"

2) There seems to be a lack of information on heterogeneity: to fully appreciate the results, I feel QQ-plots for the full GWAS, and heterogeneity measures/forest plots for the top-hits should be provided. Additionally, I would be interested to see how the effects the biobank data (locally) compare to the main analysis data (see also point 1 above).

Author response: We thank the reviewer for this suggestion, and we have now provided QQ plots and forest plots for our top signals (see results section and **Supplementary figures 1 - 3, 23, as outlined in previous comment**). In addition, we have compared the biobanks with the main GWAS in our forest plots (see our response to previous comment).

3) *At many points in the manuscript, information is lacking or only available by reading other work. Below there are some examples, but there are likely others. Specifically,*

a) the authors refer to their previous publication regarding included cohorts, classification of epilepsy - both crucial pieces of information - and in sentences like “.. with the same population prevalences as in our previous study”. This information is in supplementary table 7, why not refer to that? It would be much easier for the reader to appreciate this manuscript if the information was included.

Author response: We agree with the reviewer and thank them for this suggestion. We have now included information regarding classification of the epilepsies in the Supplementary Materials (see section titled “Classification of epilepsy”). We now also reference Supplementary Table 7 in the main manuscript, in the section entitled “heritability analysis” (page 16/17).

b) Explanation of methods is extremely brief. There is no need to repeat the papers that introduce the methodology, but the rationale for some of the analyses could be explained better. As examples: why do the authors use both LDSC and LDAK for heritability estimates and why are their differences not discussed? Why running both TWAS and SMR if they test the same hypotheses? “Summary-data-based Mendelian Randomization (SMR) v1.03 is an additional method to assess the association between epilepsy and expression of specific genes.” Another example: “..after which bivariate MiXeR was run with default settings” later in the manuscript, MiXeR is introduced as “The total amount of causally associated variants (i.e., variants with nonzero additive genetic effect) underlying epilepsy risk was calculated by a causal mixture model (MiXeR).” What is tested in such a bivariate model? “This sex difference is further corroborated by significant sex-heterogeneity ($p=1.54 \times 10^{-8}$) and gender-differentiation ($p=5.6 \times 10^{-9}$).” What exactly is tested in each of these cases? The methods section does not help.

Author response: We agree with the reviewer, and have provided a more thorough explanation of our methods and included a rationale where necessary in section Results and Methods. The rationale for using both TWAS and SMR is now described in the methods section, where the following text has been added:

“Although TWAS and SMR have similar aims, the differences in methods and reference datasets result in complementary information. As opposed to the FUSION TWAS method, which uses multi-SNP imputation

of gene expression, SMR uses Mendelian randomisation to test whether the effect size of a SNP on epilepsy is mediated by expression of specific genes.”

The rationale for using LDAK for heritability is mentioned in the methods section, which now reads:

“We calculated SNP-based heritability on the European-only GWAS using LDAK, as it was recently shown to give more accurate heritability estimates for complex traits, when compared to other methods including LDSC.^{94,95”}

We also added to the methods section, the following rationale for the genetic correlation analyses:

“Although estimates are in general consistent between LDSC and LDAK⁹⁰, we decided to use LDSC as it is the more established method of the two for genetic correlations and used by almost all genetic correlation atlases and databases.^{96,97”}

We further explained the use of the bivariate MiXeR analysis in the corresponding Methods section:

“Bivariate MiXeR analyses estimates the total amount of causal SNPs underlying each trait, after which it assesses how many of these SNPs are shared between two traits. Importantly, the amount of overlapping SNPs is calculated regardless of direction of effect. This makes it different from overall genetic correlation analyses such as LDSC, where overlapping SNPs with mixed directions of effect can cancel each other out, resulting in low genetic correlation.”

We have added the following text to the methods section of our sex-specific analyses:

“Sex-differentiated analyses are meta-analyses between female and male-only GWAS, allowing for differences between the sexes, while sex-heterogeneity tests the difference in effect size for each SNP between female-only and male-only GWAS.”

c) Supplementary Tables/Data are not readable on their own. Why not put all the data tables in one document and add an index, plus provide explanations for the column headers at the top of each table?

Author response: We thank the reviewer for this suggestion. We have now created a single Supplementary data index. In addition, we have now included explanations of all column headers in the respective files. Regarding the suggestion to merge all Supplementary data into one large Excel file, we tried but found that the amount of data tables and tabs/sheets made the file too bulky and as a result, difficult to navigate. Hence, we have decided to retain separate files for the supplementary data.

4) Drug repurposing: The conclusions on drug repurposing seem a bit circular. If done using the same procedure used by Mirza et al. (2021), then there are two parameters (gene-list length of AC score and the weight used in the AC and FM average) that are optimized for predicting known anti-epilepsy

medications (ASMs). It is understandable that the currently approved medications for epilepsy are the best treatments we have, and therefore, can be used to optimize a model. However, I don't think one can then conclude the drug repurposing results are validated based on finding ASMs using this method, since predicting ASMs has been built into the model. In addition, is a permutation test using randomly assigned scores a fair comparison, given that the ability to predict ASM scores has been built into the model? To fully appreciate the drug repurposing results, median ranks and AUROC scores of all drug groups (including ASMs) should be provided.

Author response: We are grateful to the reviewer for this comment.

We agree with the reviewer that there remains an inherent circularity when one is optimising the model for ASM and have removed the sentence that states that this was validated. Nevertheless, in our original paper of the method used here (Mizra et al), ASMs were successfully predicted using many different gene-list lengths for the AC score and using many different relative weights for the weighted mean of AC and FM scores. Although not optimal, these procedures provide a degree of validation. To extend this, we now also include an additional validation to account for some of this circularity. We performed a new analysis in which all ASMs were randomly split into two equal sets: a training set and a test set. The training set was used to optimise the drug prediction for 'all epilepsy' and the test set was used to test its performance. Results showed the performance did not decline. Specifically, when all ASMs were used to optimise drug prediction for 'all epilepsy', the average rank of all ASMs was 111, which corresponds to a percentile of 91.7, and is significantly higher than expected by chance ($p < 1 \times 10^{-6}$). When the training set of ASMs was used to optimise drug prediction for 'all epilepsy', the average rank of the test set of ASMs was 118, which corresponds to a percentile of 91.2, and is significantly higher than expected by chance ($p < 1 \times 10^{-6}$).

We have added the following sentence to the results (section entitled "Leveraging GWAS for drug repurposing", page 9) section: *"These observations were also true for a 'test set' (randomly selected 50%) of ASMs, when the remaining ASMs ('training set') were used for optimising the predictions."*

Regarding the permutation testing, we contend that it is the most fitting method to establish a test given the inherent circularity and believe that using randomly assigned scores provides a measure for significance.

Finally, median ranks and AUROCs of all drug groups (including ASMs) are now provided as a supplementary table 8, as suggested by the reviewer.

5) Results "Genomic inflation was comparable to our previous GWAS and all linkage-disequilibrium score regression (LDSR) intercepts were lower than in our previous GWAS (Supplementary table 6) suggesting that the signals are primarily driven by polygenicity, rather than by confounding or population

stratification.” At first glance, the intercept of 1.10 does not seem to be extremely high. But then (only when reading the methods) I discovered that these analyses were done in the European sample only. As a general point, it should be made clear through the manuscript (and in table/figure captions) whether the shown results stem from the trans-ancestry analyses or from the European-only analysis. In light of this, a LDSC intercept of 1.10 does not seem to exclude confounding such as population stratification. Following a commonly used approach and computing the ratio (LDSC intercept - 1) / (mean Chi2 - 1) using supplemental table 6 a substantial proportion (37%, 48% and 11%) of the inflation might be due to some form of bias (e.g. stratification, cryptic relatedness or other confounding factors). Please comment.

Author response: We thank the reviewer for this comment. We have now clarified in the text, where the analysis refers to European-only or is trans-ethnic. As suggested, we have now also detailed and discussed the attenuation rate. We have noted in the results text that part of the signal might be due to confounding or population stratification, particularly for focal epilepsy (where we found no significant locus), but less so for GGE, where we found the majority of our loci. We used mixed-model analyses which should account for cryptic relatedness. We now also compared the attenuation rates between the current GWAS, our previously published GWAS and the Epi25 (the largest additional cohort).

We have added the following text to the manuscript:

“Computation of the attenuation ratio suggested that part of the inflation signal, in particular for focal epilepsy (0.58) might be due to some form of bias (e.g. confounding or population stratification).¹⁷ The attenuation ratio was lowest for GGE (0.11), which includes the vast majority of significant loci (Supplementary table 9).”

Other points:

6) Abstract: “SNP-based heritability analyses show that common variants substantially close the missing heritability gap for GGE”. I assume this refers to the liability scale heritability but it is hard to judge without knowing the twin-heritability. In addition, in the introduction “Nevertheless, previous genetic studies of common epilepsies have explained only a few percent of this common genetic variant, or SNP-based, heritability”. What is this percentage? Please add these numbers to the manuscript.

Author response: We confirm this refers to liability scale heritability. The twin-based heritability for GGE is 0.66 (+/- 0.13). We have added the percentage explained to the abstract, as well as detailing in the introduction, the percentage points explained by previous studies.

7) Introduction, page 2: "Here, we report the third epilepsy GWAS meta-analysis..." the authors probably mean the ILAE third GWAS, there are more out there (e.g. Song et al., 2021). In addition, the caption of Figure 2 states: "Loci were classified as novel or replication according to the genome-wide significant results of previous GWAS publications" What are these publications? Related: in the results: "The selected traits had either, or a combination of 1) epilepsy as a common comorbidity or 2) pleiotropic loci shared with epilepsy". Pleiotropic loci based on what? From the methods I guess that it is the GWAS catalog?

Author response: We thank the reviewer for pointing this out. To avoid ambiguity, we now refer to the present study as the 'third GWAS by the ILAE Consortium on Complex Genetics.' Additionally we now state our definition of novel loci in Figure 1: "Novel loci were those previously unreported as GWAS-significant in previous epilepsy GWAS."

Pleiotropic loci were indeed defined based on the GWAS catalogue. This is now mentioned in the results section of the revised manuscript.

8) Results, page 2: "We further conducted meta-analyses in subjects of European ancestry of the well-defined GGE subtypes ... lesion-negative focal epilepsy". I would appreciate a mention of the sample sizes here. Were controls shared in these analyses?

Author response: We thank the reviewer for pointing this out. We now mention the sample sizes for the subphenotypes in the results section 'study overview', of the revised manuscript. Additionally, we now clearly state in the revised text, that controls were shared across the sub-phenotype analysis we present (same results section 'study overview').

9) Results, page 2: Why was functional annotation only done for the GGE analyses?

Author response: We agree with the reviewer that functional annotation of the other phenotypes would also be of relevance. We have now added functional annotations for all phenotypes to Supplementary data 1, and included an interpretation of this for the 'all epilepsy' phenotype in the main manuscript. Only CAE and JME subphenotypes presented GW-significant SNPs, but less than 300 total SNPs could be included, which is deemed as too few SNPs for any meaningful interpretation in the main manuscript. This section now reads:

*“Functional annotation of the 2,355 genome-wide significant SNPs across the 22 GGE loci and 612 SNPs from the all epilepsy loci revealed that most variants were intergenic or intronic (**Supplementary data 1**). 26/2355 (1.1%) GGE SNPs were exonic, of which 12 were located in protein-coding genes and nine were missense variants. We identified two exonic ‘all epilepsy’ SNPs, one of which (rs2298771) was a missense variant located in the gene SCN1A. Eighty-one percent of ‘all epilepsy’ SNPs and 61% of GGE SNPs were located in open chromatin regions, as indicated by a minimum chromatin state of 1-7.¹³ Further annotation by Combined Annotation-Dependent Depletion (CADD) scores predicted that 37 ‘all epilepsy’ and 110 GGE SNPs were deleterious (CADD score >12.37).”*

10) Results, page 3: “Our ‘all epilepsy’ meta-analysis revealed four genome-wide significant loci, of which two were novel (Figure 1).” The authors then move forward and discuss a sub-threshold locus. Why not discuss the location/function of the two novel loci?

Author response: We agree with the reviewer that a description of this subthreshold locus ($p=5.04 \times 10^{-8}$) seems out of place in this section. This sentence has now been removed from the manuscript and we focus on the genome-wide significant signals. We decided not to give an in-depth description of all genome-wide significant loci in this section, considering that we already do so in the paragraph ‘Locus annotation, transcriptome-wide association study (TWAS) and gene prioritisation’

11) Results, page 5: “Interrogation of the Drug Gene Interaction Database (DGIdb) showed that 13 of the 29 genes are targeted by a total of 214 currently licensed drugs (Supplementary data 6).” It would be interesting to discuss what these drugs are licensed for, and/or refer to Supplementary Table 8 (this is where I found the information, but that was not mentioned in the text until much later. Alternatively, this paragraph could be removed.

Author response: We agree with the reviewer that this information might be difficult to interpret and have now removed this section as suggested.

12) Results, page 6; Fig 2: Could the authors provide a measure of the reliability of the highest scoring genes? i.e. is this by far the highest scoring gene, or could there be other genes in this locus that score e.g. one less? In addition, are these 10 criteria truly independent - in the sense that they capture a different aspect of functionality? I can imagine with all the underlying data sources in the methods used, there might be overlap. If this is not the case, it would be good to state this explicitly in the methods section.

Author response: We agree with the reviewer that further measures of the independence of the 10 biological prioritisation criteria would be of importance to interpret these results. We have now provided the full list of all mapped genes in each locus and their corresponding biological prioritisation scores in **Supplementary data 6**. For most loci, the highest scoring gene is by far the highest scoring gene, but with such few categories it is difficult to provide an objective measure of reliability.

Additionally, we have calculated correlations between these criteria to assess the independence between the different criteria (**Supplementary table 15**). In the Methods ‘Gene prioritization’ section of our revised manuscript, we now also note that most correlations between criteria were low (range: -0.13 to 0.39), suggesting that they convey complementary information.

This section now reads: *“The gene with the highest score was chosen as the most likely implicated gene (see **Supplementary data 6** for a complete list of scores for all genes in each locus) We implicated both genes if they had an identical, highest score. We calculated Pearson correlation coefficients between the 10 loci (**Supplementary table 15**) and note that most correlations were low (range: -0.13 to 0.39), suggesting that they convey complementary information.”*

13) *Supplementary Data 5: why are there NAs in the table?*

Author response: The column “P_heidi” denotes “NA “when fewer than 3 SNPs for the HEIDI heterogeneity test are available, so no heterogeneity P-value can be calculated. This is now detailed in the legend of Supplementary data 5.

14) *Results, page 8: The rationale for the analysis between ASD, intelligence and epilepsies is not clear (or rather, receives a lot of attention compared to other parts). If I understand the reasoning correctly, the authors attempt to show that the percentage of discordant/concordant directions of effect are similar associating epilepsy with either ASD or intelligence. This could very well be, but it does not explain why intelligence then shows a global association and ASD does not. MiXeR provides a different type of results compared to LDSC, so I do not see why such an comparison would imply overlap between ASD and GGE. Perhaps the authors wish to run the MiXeR analyses for all phenotypes and compare results; then the difference in interpretation between LDSC and MiXeR analyses should be discussed. In addition, couldn't the lack of correlation be a consequence of the limited sample size for ASD?*

Author response: We agree that the rationale for specifically assessing ASD and intelligence in relation to epilepsy with MiXeR was unclear. As suggested, we performed MiXeR analyses for all 18 phenotypes (the same as assessed with LDSC) and now provide a more complete analysis and concluding statement, which reads:

“For most selected traits, the direction of effect was concordant for 40-60% of SNPs. This might explain why some LDSC correlations were low, together with other relevant factors including sample size, polygenicity, and trait genetic architecture.”

15) Discussion, page 10: *“It would seem that focal epilepsies, as a group, are far more heterogeneous than GGE”. Could an alternative explanation be that this phenotype is more polygenic?*

Author response: We thank the reviewer for giving us the opportunity to further discuss this important point. We agree that more explanations are equally likely and have now added a sentence to the discussion that now reads:

“In contrast to GGE, for focal epilepsies we found only a minor contribution of common variants, with no variant reaching genome-wide significance. It would seem that focal epilepsies, as a group, are far more heterogeneous than GGE, lack loci with high effect sizes, have a higher degree of polygenicity, and/or have a lower contribution of common heritable risk variation.”

16) Methods, page 12: *Why the difference in QC criteria for the new cohorts and Epi25? Specifically, the difference in INFO score cut-off is striking. As a side note, methods are describing QC procedures with either filtering criteria or “QCed for” and these criteria have opposite signs, which can be confusing. Please harmonise.*

Author response: We agree with the reviewer that our previous description of the genotyping and quality control was confusing. The Methods paragraph entitled ‘Genotyping, quality control and imputation’ has now been rewritten to harmonise our descriptions. We have now also explained why there were differences in QC between the Epi25 and other cohorts.

17) Methods, page 15: *“Brain expression: we calculated mean expression of all brain and non-brain tissues based on data from the Genotype-Tissue Expression (GTEx) project v8 and assessed if the average brain tissue expression was higher than the average expression in non-brain tissues.” This criterion is not clear to me as it is not mentioned how the GWAS results come into this. Is it tissue-specificity - similar to tissue-specificity implemented in FUMA?*

Author response: For this criterion we used the GTEx database to assess whether each of the mapped genes is preferentially expressed in the brain, as compared to other organs/tissues in the body, under the hypothesis that genes expressed in the brain are more likely to be involved in epilepsy. Note this is different from tissue specificity analyses implemented in FUMA which utilises genome-wide gene based

association study (GWAS) results to compute whether the strength of GWAS association is correlated with expression levels of genes in certain tissues. We have now expanded upon this section in the methods of our manuscript, which now reads:

“Brain expression: for each mapped gene we calculated the mean expression in all brain and non-brain tissues based on data from the Genotype-Tissue Expression (GTEx) project v8⁸⁶. Next, we assessed whether the gene was more strongly expressed in brain tissues than non-brain tissues, by comparing the average expression in all brain tissues with all non-brain tissues.”

Reviewer #2:

Remarks to the Author:

The authors present updated results from GWASs in epilepsy subtypes with secondary analyses further characterizing their findings and their potential implications regarding clinical ramifications (e.g. by repurposing existing drugs). The presented analyses and results are certainly interesting and well described (albeit some sections would benefit from slightly more details). However, the biggest criticism (see below) can probably be summarized as "missed opportunities" as current GWAS papers in complex traits genetics often include more analytical approaches (especially when it comes to heterogeneous phenotypes such as the ones presented here). Without these additional analyses, this is still a remarkable effort and manuscript but likely less interesting to a broader readership. It obviously can't be ruled out that such analyses are earmarked for additional manuscripts, but this reviewer would support inclusion in the current manuscript.

Major points:

(1) Results from the presented analyses (and previous GWAS) indicate heterogeneity in the different epilepsy subtypes when it comes to the underlying genetic architecture. The authors take only modest interest in analyzing the heterogeneity given the results. Granted, the low sample size might fuel this in the different sub-types but at least for some of their sub-types sample size seems high enough to warrant such an analysis. For example, a case-case GWAS between FE and GGE could help shed light on the difference in genetic architecture between the two. Applying tools such as GW-SEM (if individual-level data is available across datasets) or at least genomic SEM would potentially provide further insights (e.g., concerning a common genetic factor across subtypes). Given the heterogeneity, it might have also made sense to use such tools to perform an overall analysis (with presumably added power compared to the crude meta-analysis for the "all epilepsy" analysis). Tools such as MTAG could have helped (with obvious limitation on small sample size, see above) to increase the "virtual" sample size for their subtypes. Tools such as METASOFT and their package to produce P-M plots could have helped to characterize the heterogeneity (<https://journals.plos.org/plosgenetics/article?id=10.1371/journal.pgen.1002555>). Again, it is

appreciated that sample size might be an issue here for some of the subtypes, however, for others, it is not and some of the mentioned tools (e.g. METASOFT) could have helped to answer questions around sample size limitations.

Author response: We thank the reviewer for providing these interesting and helpful suggestions to further improve our manuscript. We have now conducted an extensive analysis of the heterogeneity reported in the original submission.

Specifically we have:

1) Created forest plots and P-M plots to characterise heterogeneity between the GGE subphenotypes and show that the GGE top SNPs are almost all homogeneously driven by all four GGE subphenotypes (**Supplementary figures 2 & 3**).

2) Deployed Genomic SEM as suggested, results from which confirm the strong correlations between GGE subtypes, but also the presence of subtype specific signals (see Supplementary Figure 25). We have added the following sentence to the main text of section ‘SNP-based heritabilities’:

*“Multivariate modelling of genetic correlation using Genomic SEM²⁹ confirmed that most of the heritability signal is shared amongst the four GGE syndromes, with some subtype specific signal (**Supplementary figure 25**).”*

3) As suggested by the reviewer, we have now also applied MTAG to leverage the strong correlations between epilepsy subphenotypes (**Supplementary table 8; Supplementary figure 11**) and, for GGE, to boost the effective sample size of individual GGE subphenotypes. However, the analysis showed concordance with the main analysis, showing significance only for the loci already identified in the main GWAS analysis. Similarly for the GGE subtypes the results showed concordance with the main analysis. To describe these new confirmatory results we have added the following sentences to the main text:

*“We applied MTAG¹⁷ to exploit the correlation between FE and GGE, boosting the effective sample size. Results were concordant with our main analysis and novel signals did not emerge (**Supplementary figure 26**).”*

“MTAG¹⁶ analysis of individual GGE subphenotypes showed concordance with the main GGE GWAS, without identifying novel loci. In addition, this analysis confirmed that the majority of GWAS significant SNPs in GGE are overlapping”(Supplementary figures 6-7).”

(2) Along the lines of the comment above the relationship between the currently presented samples (and

results) in the main analyses and biobanks should probably be further characterized. It is by now a well-known observation that different ascertainment schemes (especially in the context of incidence vs. prevalence samples etc.) lead to different “cuts” of the phenotypic spectrum of a disorder being on display and, as such, differences in genetic architecture being identified (with often misaligned heritability figures, etc.). At a bare minimum, the currently presented observation would point to potential ramifications for the interpretation of their results in the main analyses (and generalizability).

Author response: We agree that the different ascertainment schemes underlying the cohorts included in our analysis are probably presenting different “cuts of the phenotypic spectrum” as the reviewer suggests.

In an effort to shed further light on the consistency of signal across the different cohorts included, we now provide forest plots, comparing 1) the biobanks, 2) ILAE_2, 3) EPI25 and 4) the current main GWAS meta analyses, ILAE_3 (see results section and **Supplementary figures 3 and 23**).

These forest plots clearly illustrate consistency in direction of effect in the biobanks for all but one top SNP in the main all epilepsy (**Supplementary figure 23**) and GGE meta-analyses (**Supplementary figure 3**), albeit with smaller effect sizes.

We have added these results to the section describing the Biobank results, which now reads:

*“Forest plots showed a consistent direction of effect between the biobanks and our primary GWAS for all biobank-genotyped genome-wide significant top SNPs of the ‘all epilepsy’ GWAS and for all but one GGE top SNP (**Supplementary figures 3 and 23**).”*

These results further confirm the strong, but incomplete, genetic correlations between the biobanks and our main GWAS (**Supplementary table 14**).

Considered together, we feel these additional analysis/results suggest that the same genetic risk/protection variants underlie the biobank and main epilepsy GWAS, but the signal in the biobanks is weaker. We feel these additional analyses further support our hypothesis stated in the discussion on how the weaker biobank signal could be due to ascertainment in biobanks for milder epilepsy, combined with less well characterised and more heterogeneous phenotypes in the biobanks (in effect the ramifications the reviewer refers to in their comment).

(3) In the interpretation and analyses of genetic correlations between autism spectrum disorders (ASD), intelligence, and epilepsy sub-types, the authors emphasise the observation that despite low r_G between ASD and epilepsy subtypes, there could be potential overlap between the genetic architecture for both conditions. This is important to note but falls short of further issues: among others, the shared genetic architecture between epilepsy subtypes and ASD might only reflect overlap with parts of the phenotypic

spectrum of the condition (i.e., the part that concerns intellectual disability). Furthermore, other aspects of the genetic correlation with the traits in Figure 3 went unstudied (in this comparison, for example, the relationship between ADHD, ASD, intelligence/cognition/EA, and epilepsy subtypes (with links between ADHD and ASD and stronger r_G between epilepsy subtypes and ADHD compared to ASD). A more detailed characterization of the genetic relationship with febrile seizures and, in general, with phenotypes for which there seems to be a discordance (either because of sample size or real genetic differences) in r_G for epilepsy subtypes would likely have helped to further characterise the genetic architecture for epilepsy. This could, for example, also have included LAVA-like analyses (i.e., analyses of regional r_G).

Author response: We thank the reviewer for these suggestions and we have now run additional Mixer and LAVA analyses. We have rewritten the section on genetic correlations, to clarify our approach, and have extended our Mixer analysis, and do not discuss the low r_G with ASD.

This section now reads: “Genetic correlation analyses assess the aggregate of shared genetic variants associated with two phenotypes. However, genetic correlations can become close to zero when there is inverse directionality of SNP effects between two phenotypes.³⁸ To explore this further, we applied MiXer to quantify polygenic overlap between GGE and the same 18 selected traits, irrespective of genetic correlation (see Methods). Results showed a large polygenic overlap between epilepsy and various other polygenic brain traits (**Supplementary Figure 20**). For most selected traits, the direction of effect was concordant for 40-60% of SNPs. This might explain why some LDSC correlations were low, together with other relevant factors including sample size, polygenicity, and trait genetic architecture. “

Second, we have performed LAVA analysis to further explore the genetic relationship with febrile seizures, ASD and intelligence. However, for febrile seizures, the results show regional significance for only one region, which includes the *SCN1A* locus. Therefore, with the current dataset, LAVA has very limited value for characterising the genetic architecture to explain concordance and discordance of global r_G . To illustrate, we found 9 regions that showed joint effects for FE and GGE, one for febrile seizures and each epilepsy subtype, none for ASD, and one (again, the *SCN1A* region) for intelligence. We believe that our study is currently still underpowered to perform regional genetic correlation analysis for joint effects between more than one trait, and therefore decided to exclude these LAVA analyses in the revision.

(4) Sex-stratified analyses have the inherent feature of reducing sample size significantly. While that might not lead to differences in reduction of sample size for the two sexes (i.e., their GWAS sample size are the same), it still opens the possibility of random fluctuation in GWAS results. This is even likely at lower sample sizes that only capture part of the genetic architecture of a given trait. To truly identify loci that have different influences on the genetic architecture, one would have to use different tools or

approaches. Case-case analyses (see above) might have helped highlight genetic architecture differences. Additionally, random subsampling of the sample (i.e., drawing random samples at the same sample size as stratified GWASs) might have added insights and helped answer the question of whether some loci are associated with one sex but not the other.

Author response: We agree with the reviewer that sex stratified analyses have the limitation of reducing sample size, with the risk of some random fluctuation in results. We have now mentioned this limitation in our results section ('Sex-specific analyses') through the following sentence:

"These analyses were limited by a reduction in sample size and prone to random fluctuation."

We have performed sex-stratified and sex-heterogeneity tests and genetic correlations between the sexes. These analyses show that the genetics of males and females are largely similar, with the exception of one putative sex-divergent signal. We note that these analyses were exploratory in nature, and we are planning to perform further in-depth analyses of potential sex-related differences in a later manuscript.

(5) When comparing current results to their previous GWAS, the authors generally only conclude that they identified additional loci. However, they fail to assess whether they also "lost" loci. It is to be supported that they make efforts to refine their previous findings (e.g., when it comes to the likely causal genes). Still, it seems equally important to highlight which associations are no longer supported. I might have overlooked this, but what happened, for example, to the association on chromosome 4 close to GABRA2 for GGE or focal epilepsy finding on chromosome 2? The latter is still present in the "all epilepsy" analysis. Still, these differences might point to changes in the phenotypic spectrum present in the samples, and analyses comparing results from previous samples and the new ones added here could help interpret the results. That might have been published elsewhere, but a small recap might be helpful.

Author response: We agree with the reviewer that an assessment of non-replicated loci would be of importance in the interpretation of our results. The large majority of loci found in our previous (2018) GWAS were replicated in the current analysis. In response to the reviewer comment, we now added a description of nonreplication in the manuscript and added a table that contains the comparison. This includes the 4p12 (*GABRA2*) locus, which is no longer genome-wide significant in the overall GGE analysis, but appears to be specific to the GGE subsyndrome, JME. Similarly, the signal at 2q24.3 is not replicated at genome-wide significance for our focal epilepsy analysis, but remains genome-wide significant in the 'all epilepsy' and GGE analysis. Across the different phenotypes, six previously reported loci were not replicated at genome-wide significance level, but their association is not completely lost (p-values ranged between 4.28E-5 and 5.35E-8).

We have therefore added the following sentence to the result section:

*“The vast majority of loci reported in our previous effort⁴ remained GWAS significant. A summary of loci that fell below genome-wide significance threshold is provided in **Supplementary table 16.**”*

Minor points:

(1) The authors claim (page 8, sex-specific analysis) that two lead SNPs from their analyses are not in LD (rs72845653, rs11191156). They base this observation on a r^2 value of 0.05. Unfortunately, they fail to appreciate that at the same time D' is at 0.87, and hence the markers are, in fact, in LD, albeit not in perfect LD (which is also not expected given the difference in MAF). See also here: https://ldlink.nci.nih.gov/?var1=rs72845653&var2=rs11191156&pop=CEU%2BTSI%2BFIN%2BGBR%2BIBS&genome_build=grch37&tab=ldpair).

Author response: We thank the reviewer for pointing this out. We have now added the D' value to the manuscript text, and we clarified that these SNPs show low allelic correlation LD.

(2) It seems that the authors used analytical tools (e.g., LDSC, MAGMA, etc.) that require reference samples. However, in some cases, there seems to be no information available about which reference samples were used. For the instances where it was denoted (for the LDAK-based heritability analyses), it might have been interesting to see what different reference samples, including representation from regions that have been present in the study sample, would have yielded (given the heterogeneity of the study sample with samples from Finland, Norway, Austria, Switzerland, etc. with sometimes significant differences in contribution to the different subtype GWASs).

Author response: We now note in the methods section, the reference samples used throughout the manuscript. In general, we restricted analyses which were dependent on LD-structure to European samples and default European reference samples for the different analyses. We realise that there is heterogeneity within our European-only dataset, however, we feel that a comprehensive analysis of the various available reference samples falls beyond the scope of the current manuscript.

Decision Letter, first revision:

27th February 2023

Dear Sam,

Your revised Article "Genome-wide meta-analysis of over 29,000 people with epilepsy reveals 26 loci and subtype-specific genetic architecture" has been seen by the original referees. You will see from their comments below that, while they find the work improved, they have raised a few ongoing technical concerns. We remain interested in the possibility of publishing your study in Nature Genetics, but we would like to consider your response to these remaining concerns in the form of a further revision before we make a final decision on publication.

As before, to guide the scope of the revisions, the editors discuss the referee reports in detail within the team, including with the chief editor, with a view to identifying key priorities that should be addressed in revision, and sometimes overruling referee requests that are deemed beyond the scope of the current study. In this case, we ask that you examine the issue of effect size heterogeneity across the different study strata and carefully address whether the individual studies contributing to the meta-analysis have been correctly weighted. We again hope you will find this prioritized set of referee points to be useful when revising your study. Please do not hesitate to get in touch if you would like to discuss these issues further.

We therefore invite you to revise your manuscript again taking into account all reviewer and editor comments. Please highlight all changes in the manuscript text file. At this stage, we will need you to upload a copy of the manuscript in MS Word .docx or similar editable format.

*2) If you have not done so already please begin to revise your manuscript so that it conforms to our Article format instructions, available http://www.nature.com/ng/authors/article_types/index.html here. Refer also to any guidelines provided in this letter.

Please be aware of our <https://www.nature.com/nature-research/editorial-policies/image-integrity> guidelines on digital image standards.

[redacted]

We hope to receive your revised manuscript within 4-8 weeks. If you cannot send it within this time, please let us know.

Sincerely,
Kyle

Kyle Vogan, PhD
Senior Editor
Nature Genetics
<https://orcid.org/0000-0001-9565-9665>

Referee expertise:

Referee #1: Genetics, neurology

Referee #2: Genetics, neurodevelopmental disorders, psychiatric diseases

Reviewers' Comments:

Reviewer #1:
Remarks to the Author:

The authors made additions to the manuscript that generally improved transparency and credibility. The answer to one of my questions, however, raised a new issue which I feel should be addressed:

Considering the lack of replication (or lack of increasing signal when adding the biobank data), the authors added (a.o.) a forest plot (Supplementary Fig 3) showing that, for their top findings, the biobank derived effects are generally in the same direction but less strong as the main ILAE3 data, which they summarized as follows: "Considered together, we feel these additional analysis/results suggest that the same genetic risk/protection variants underlie the biobank and main epilepsy GWAS, but the signal in the biobanks is weaker." Looking at Supplementary Figure 3, I agree with this statement when comparing ILAE2/3 with the biobank data. However, what this figure also shows, in my opinion, is that the boost in findings when comparing ILAE2 and ILAE3 might be completely caused by the Epi25 cohort(s). Do the authors have an explanation of this? With the Epi25 cohort being a cohort that was analysed separately, there are so many differences (QC, GWAS-analysis software, ancestry) that it is hard to judge where this difference came from. The difference in effect size is so striking that I even wonder whether - given all the aforementioned differences - the effect sizes can be directly compared. Please comment and discuss this heterogeneity in the manuscript.

Other points:

- I do appreciate the changes made to the manuscript regarding the drug repurposing, but I am afraid that while the additions to the manuscript might be clear to the reviewers, it is not clear to the general audience. For example, the statement "These observations were also true for a 'test set' (randomly selected 50%) of ASMs, when the remaining ASMs ('training set') were used for optimising the predictions." is not at all clear; please add a rationale (i.e. the original method was targeted at ASMs and the model was retrained for the purpose of this manuscript) and short description to the methods. To fully appreciate the approach, it would be helpful to also add the sample size of each drug group, and the training-testing split results to the supplementary data file.

- Please check the numbering of the supplementary material; it might be the editorial system, but some files seem to be misnumbered.

- Supplementary Figure 11: If I understand correctly, this figure combines the genetic correlations (from LDSC) with heritability measures from LDKA, but the latter is not mentioned in the caption. It might be useful to add, since now there is a statement "# rg out of bounds due to phenotype not reaching significant heritability" next to a heritability estimate of 25% (19%-31%).

Reviewer #2:

Remarks to the Author:

Thank you for allowing me to review the revised manuscript "Genome-wide meta-analysis of over 29,000 people with epilepsy reveals 26 loci and subtype-specific genetic architecture". Adding to the already impressive body of work as foundation in the previous version, the authors have addressed almost all comments from the two reviewers. As a result the manuscript has been significantly improved. As before, this work represents a crucial collaborative effort to increase sample size for identification of genetic loci associated with epilepsy (including gene prioritization, functional annotations, and other secondary analysis to characterize their findings). I sincerely appreciate the effort the authors undertook to address all reviewer comments and happily accept that not all comments led to new valuable insights (e.g., concerning the LAVA and MTAG analyses), which

probably was due to the limited utility of the suggestion to begin with. I am also happy with their response to the comment for the sex-stratified analyses and look forward to reading their manuscript on this topic.

While reading through the manuscript again, I stumbled across a comment that I should have made before, which concerns the SAIGE analyses and the use of effective sample sizes for the meta-analyses. By design and nature SAIGE analysis results do not reflect effective sample sizes when there are huge differences in case and control numbers, which is one of the use cases for SAIGE (i.e., large-scale biobank PheWASs with low case numbers and large control number). As a consequence the actual effective sample size of such analyses are lower than what would be expected from the formula that the authors used (and, as such, the weight for these studies in the meta-analyses would be too high). SAIGE (at least in some versions) also seems to have an issue with inaccurate SEs (and differences compared to, for example REGENIE). At this stage I am not sure how to deal with a comment like that, but I would suggest that the authors have one last look at their analyses to see whether any of these problems could have affected their results. I am genuinely sorry for not bringing this up earlier.

Before going into production, I would suggest spending some time enhancing the supplementary information in terms of consistency of formatting (e.g., same font and font size across different sections / tables / figures).

Author Rebuttal, first revision:

Point by point response to reviewers

6/3/23

Reviewers' Comments:

Reviewer #1:

Remarks to the Author:

The authors made additions to the manuscript that generally improved transparency and credibility. The answer to one of my questions, however, raised a new issue which I feel should be addressed:

Considering the lack of replication (or lack of increasing signal when adding the biobank data), the authors added (a.o.) a forest plot (Supplementary Fig 3) showing that, for their top findings, the biobank derived effects are generally in the same direction but less strong as the main ILAE3 data, which they summarized as follows: "Considered together, we feel these additional analysis/results suggest that the same genetic risk/protection variants underlie the biobank and main epilepsy GWAS, but the signal in the biobanks is weaker." Looking at Supplementary Figure 3, I agree with this statement when comparing ILAE2/3 with the biobank data. However, what this figure also shows, in my opinion, is that the boost in

findings when comparing ILAE2 and ILAE3 might be completely caused by the Epi25 cohort(s). Do the authors have an explanation of this? With the Epi25 cohort being a cohort that was analysed separately, there are so many differences (QC, GWAS-analysis software, ancestry) that it is hard to judge where this difference came from. The difference in effect size is so striking that I even wonder whether - given all the aforementioned differences - the effect sizes can be directly compared. Please comment and discuss this heterogeneity in the manuscript.

Author response:

We thank the reviewer for their continued careful reading and constructive comments on the manuscript. The reviewer is correct that most of the boost in findings from ILAE2 to ILAE3 is due to Epi25 which contributed the majority of the new cases. Whilst there were some differences between ILAE2 and Epi25 with respect to QC, GWAS-analysis software, ancestry etc we were reassured that the findings seen initially in ILAE2 were replicated in the Epi25 data and that the directionality of the new significant SNPs in ILAE3 was the same in ILAE2 (Supplementary figure 22 & 23; note that numbering has been updated). Regarding the Biobank data, we were reassured that for ILAE3 significant SNPs, the direction of the effect was the same in the Biobanks (Supplementary figure 22 & 23), although the Biobanks did not increase the overall effect sizes, nor reveal much in way of novel findings. We believe this is due to the lower phenotyping quality in Biobanks as well as increased heterogeneity of the epilepsy samples included.

The reviewer has correctly highlighted an issue with comparing the effect sizes shown in Supplementary figure 22 & 23. The apparent differences in effect sizes shown in these figures are a scaling artefact in the creation of these forest plots. Beta and standard error (SE) coefficients for each cohort are required to create forest plots. However, only Z-scores and P-values were available for the ILAE2 and ILAE3 meta-analyses, as we had performed P-value based meta-analyses which does not incorporate beta and SE values. We therefore used the following formulas to calculate an approximation of beta and SE coefficients for ILAE2 and ILAE3:

$$^1 \text{Beta} = z / \sqrt{2p(1-p)(n + z^2)} \text{ and } \text{SE} = 1 / \sqrt{2p(1-p)(n + z^2)}.$$

Supplementary figures 22 & 23 illustrate how the scales of this conversion correspond poorly with those of the beta and SE of the Epi25 cohort, for which beta and SE coefficients were available from the current GWAS. The ILAE2 and ILAE3 beta and SE values are proportionately lower relative to the Epi25 cohort. Due to this scaling artefact, the effect sizes should not be directly compared between the different cohorts, but consistency in direction of effect can. Although we appreciate that the scaling might appear confusing, we decided to retain this figure given that the main purpose was to compare the direction of effect between the current main GWAS and the Biobank GWAS.

We have altered the legend to Suppl Fig 22 to highlight this scaling artefact and to emphasize the comparison of directionality of effects, rather than of effect sizes. It is important to note that this conversion was not used for any of the main analyses/GWAS in this manuscript. The main GWAS constituted a P-value based meta-analyses in METAL, weighted by effective sample size (see Methods), for which no beta or SE are required. As such, the Epi25 and ILAE2 cohorts contributed roughly equally to the current ILAE3 analyses, since sample sizes were comparable.

the legend for supplementary figure 22 now reads:

“Supplementary figure 22: Consistency of directionality of genome-wide significant SNPs for GGE, CAE and JME in ILAE3 (this study) with ILAE2, Epi25 data and Biobanks. We show forest plots of the GGE, CAE and JME top hits showing effect sizes (Beta) and standard errors (SE) for each of the sumstats. Note that the directionality of the effects are largely consistent but the magnitude of effect sizes should not be directly compared between cohorts due to scaling differences. For ILAE2 and ILAE3 meta-analysis, beta and SE values were approximated from Z-scores and P-values as described previously.²³ The scaling of this conversion does not directly match the beta and SE values of the Epi25 and Biobank GWAS. The data from 7 SNPs are not shown for the Biobank cohort due to missing genotype data. Genome-wide significant SNPs are displayed as filled points and non-significant as hollow points. X-axis limits are truncated between (-0.20,+0.20). “

Other points:

- I do appreciate the changes made to the manuscript regarding the drug repurposing, but I am afraid that while the additions to the manuscript might be clear to the reviewers, it is not clear to the general audience. For example, the statement "These observations were also true for a 'test set' (randomly selected 50%) of ASMs, when the remaining ASMs ('training set') were used for optimising the predictions." is not at all clear; please add a rationale (i.e. the original method was targeted at ASMs and the model was retrained for the purpose of this manuscript) and short description to the methods. To fully appreciate the approach, it would be helpful to also add the sample size of each drug group, and the training-testing split results to the supplementary data file.

Author response:

We thank the reviewer for this comment. To help clarify the text, we have added to the methods section, a rationale and methodology for the validation analysis. We have also added results for the validation step to the Supplementary note. These are referred to in the results section of the main text.

Furthermore, as requested we have included the sample size of each drug group in Supplementary data 8.

the following text has been added to the Methods section of the manuscript:

“Given the complete set of all ASMs was used to optimise these predictions,³⁹ for further validation, ASMs were randomly split into two sets of equal size: a training set to optimise the drug prediction and a test set to assess performance.”

In the manuscript Results text refers to the methods above, and to the Supplementary Note which now includes the following:

“To enable optimal objective assessment of the methodology for predicting ASMs, we created a comprehensive list of effective drugs used currently or previously for the treatment of epilepsy or seizures in people, as previously described.²⁵ ASMs were randomly split into two sets of equal number: a training set and a test set. The training set was used to optimise the drug prediction for all epilepsy and the test set was used to test its performance. When all ASMs were used to optimise drug prediction for all epilepsy, the average rank of all ASMs was 111, which corresponds to a percentile of 91.7, and is significantly higher than expected by chance ($p < 1 \times 10^{-6}$). When the training set of ASMs was used to optimise drug prediction for all epilepsy, the average rank of the ASMs in the test set was 118, which corresponds to a percentile of 91.2, and is significantly higher than expected by chance ($p < 1 \times 10^{-6}$).”

- Please check the numbering of the supplementary material; it might be the editorial system, but some files seem to be misnumbered.

Author response:

We thank the reviewer for pointing this out. We have now corrected the errors in the numbering of supplementary materials.

- Supplementary Figure 11: If I understand correctly, this figure combines the genetic correlations (from LDSC) with heritability measures from LDAK, but the latter is not mentioned in the caption. It might be useful to add, since now there is a statement “# rg out of bounds due to phenotype not reaching significant heritability” next to a heritability estimate of 25% (19%-31%).

Author response:

We agree with the reviewer and have now added the suggested detail to the legend of **Supplementary figure 11**.

Reviewer #2:**Remarks to the Author:**

Thank you for allowing me to review the revised manuscript "Genome-wide meta-analysis of over 29,000 people with epilepsy reveals 26 loci and subtype-specific genetic architecture". Adding to the already impressive body of work as foundation in the previous version, the authors have addressed almost all comments from the two reviewers. As a result the manuscript has been significantly improved. As before, this work represents a crucial collaborative effort to increase sample size for identification of genetic loci associated with epilepsy (including gene prioritization, functional annotations, and other secondary analysis to characterize their findings). I sincerely appreciate the effort the authors undertook to address all reviewer comments and happily accept that not all comments led to new valuable insights (e.g., concerning the LAVA and MTAG analyses), which probably was due to the limited utility of the suggestion to begin with. I am also happy with their response to the comment for the sex-stratified analyses and look forward to reading their manuscript on this topic.

While reading through the manuscript again, I stumbled across a comment that I should have made before, which concerns the SAIGE analyses and the use of effective sample sizes for the meta-analyses. By design and nature SAIGE analysis results do not reflect effective sample sizes when there are huge differences in case and control numbers, which is one of the use cases for SAIGE (i.e., large-scale biobank PheWASs with low case numbers and large control number). As a consequence the actual effective sample size of such analyses are lower than what would be expected from the formula that the authors used (and, as such, the weight for these studies in the meta-analyses would be too high). SAIGE (at least in some versions) also seems to have an issue with inaccurate SEs (and differences compared to, for example REGENIE). At this stage I am not sure how to deal with a comment like that, but I would suggest that the authors have one last look at their analyses to see whether any of these problems could have affected their results. I am genuinely sorry for not bringing this up earlier.

Author response:

We thank the reviewer for their ongoing positive and constructive appraisal of the manuscript. We agree it is important to accurately weight the different cohorts contributing to the meta-analyses. For the Epi25 analyses we used SAIGE, which calibrates results when analysing cohorts with extreme case-control imbalance, as can be the case with population biobanks. SAIGE better controls for such

imbalance, relative to linear- and logistic mixed models.² The Epi25 cohort included 11,544 epilepsy cases and 13,121 controls. Since the number of cases and controls is similarly matched, we feel confident that the 'effective sample size' weighting is comparable between the different cohorts included in our meta-analyses. Furthermore, we performed P-value based meta-analyses which doesn't utilize beta and SE values. As such, we are confident our final results are unaffected by any potential inaccuracies in the scaling/calculation of beta and SE values.

Before going into production, I would suggest spending some time enhancing the supplementary information in terms of consistency of formatting (e.g., same font and font size across different sections / tables / figures).

Author response:

We have now updated the revised supplementary materials for consistency in formatting.

References

1. Zhu, Z. *et al.* Integration of summary data from GWAS and eQTL studies predicts complex trait gene targets. *Nat. Genet.* **48**, 481–487 (2016).
2. Zhou, W. *et al.* Efficiently controlling for case-control imbalance and sample relatedness in large-scale genetic association studies. *Nat. Genet.* **50**, 1335–1341 (2018).

Decision Letter, second revision:

Dear Dr. Berkovic,

Thank you for submitting your revised manuscript "Genome-wide meta-analysis of over 29,000 people with epilepsy reveals 26 loci and subtype-specific genetic architecture" (NG-A60262R1). In light of your responses to the points raised at the previous round of review, we will be happy in principle to publish your study in Nature Genetics as an Article pending final revisions to comply with our editorial and formatting guidelines.

We are now performing detailed checks on your paper, and we will send you a checklist detailing our editorial and formatting requirements soon. Please do not upload the final materials or make any revisions until you receive this additional information from us.

Thank you again for your interest in Nature Genetics. Please do not hesitate to contact me if you have any questions.

Sincerely,
Kyle

Kyle Vogan, PhD
Senior Editor
Nature Genetics
<https://orcid.org/0000-0001-9565-9665>

Final Decision Letter:

21st July 2023

Dear Dr. Berkovic,

I am delighted to say that your manuscript "Genome-wide association meta-analysis of over 29,000 people with epilepsy identifies 26 risk loci and subtype-specific genetic architecture" has been accepted for publication in an upcoming issue of Nature Genetics.

Your paper will be published online after we receive your corrections and will appear in print in the next available issue. You can find out your date of online publication by contacting the Nature Press Office (press@nature.com) after sending your e-proof corrections. Now is the time to inform your Public Relations or Press Office about your paper, as they might be interested in promoting its publication. This will allow them time to prepare an accurate and satisfactory press release. Include your manuscript tracking number (NG-A60262R2) and the name of the journal, which they will need when they contact our Press Office.

Before your paper is published online, we will be distributing a press release to news organizations

worldwide, which may very well include details of your work. We are happy for your institution or funding agency to prepare its own press release, but it must mention the embargo date and Nature Genetics. Our Press Office may contact you closer to the time of publication, but if you or your Press Office have any enquiries in the meantime, please contact press@nature.com.

Please note that Nature Genetics is a Transformative Journal (TJ). Authors may publish their research with us through the traditional subscription access route or make their paper immediately open access through payment of an article-processing charge (APC). Authors will not be required to make a final decision about access to their article until it has been accepted. [Find out more about Transformative Journals](https://www.springernature.com/gp/open-research/transformative-journals)

Authors may need to take specific actions to achieve [compliance](https://www.springernature.com/gp/open-research/funding/policy-compliance-faqs) with funder and institutional open access mandates. If your research is supported by a funder that requires immediate open access (e.g. according to [Plan S principles](https://www.springernature.com/gp/open-research/plan-s-compliance)) then you should select the gold OA route, and we will direct you to the compliant route where possible. For authors selecting the subscription publication route, the journal's standard licensing terms will need to be accepted, including [self-archiving-and-license-to-publish](https://www.nature.com/nature-portfolio/editorial-policies/self-archiving-and-license-to-publish). Those licensing terms will supersede any other terms that the author or any third party may assert apply to any version of the manuscript.

Please note that Nature Portfolio offers an immediate open access option only for papers that were first submitted after 1 January 2021.

If you have not already done so, we invite you to upload the step-by-step protocols used in this manuscript to the Protocols Exchange, part of our on-line web resource, natureprotocols.com. If you complete the upload by the time you receive your manuscript proofs, we can insert links in your article that lead directly to the protocol details. Your protocol will be made freely available upon publication of your paper. By participating in natureprotocols.com, you are enabling researchers to more readily reproduce or adapt the methodology you use. [Natureprotocols.com](https://natureprotocols.com) is fully searchable, providing your protocols and paper with increased utility and visibility. Please submit your protocol to <https://protocolexchange.researchsquare.com/>. After entering your [nature.com](https://www.nature.com) username and password you will need to enter your manuscript number (NG-A60262R2). Further information can be found at <https://www.nature.com/nature-portfolio/editorial-policies/reporting-standards#protocols>

Sincerely,
Kyle

Kyle Vogan, PhD
Senior Editor
Nature Genetics
<https://orcid.org/0000-0001-9565-9665>